# Activation and allosteric regulation of the orphan GPR88-Gi1 signaling complex

Geng Chen[1,2,7], Jun Xu[1,3,7], Asuka Inoue [4,7✉], Maximilian F. Schmidt [5,7], Chen Bai[6], Qiuyuan Lu[1], Peter Gmeiner [5✉], Zheng Liu [1✉] & Yang Du [1✉]

GPR88 is an orphan class A G-protein-coupled receptor that is highly expressed in the striatum and regulates diverse brain and behavioral functions. Here we present cryo-EM structures of the human GPR88-Gi1 signaling complex with or without a synthetic agonist *(1R, 2R)*-2-PCCA. We show that *(1R, 2R)*-2-PCCA is an allosteric modulator binding to a herein identified pocket formed by the cytoplasmic ends of transmembrane segments 5, 6, and the extreme C terminus of the α5 helix of Gi1. We also identify an electron density in the extracellular orthosteric site that may represent a putative endogenous ligand of GPR88. These structures, together with mutagenesis studies and an inactive state model obtained from metadynamics simulations, reveal a unique activation mechanism for GPR88 with a set of distinctive structure features and a water-mediated polar network. Overall, our results provide a structural framework for understanding the ligand binding, activation and signaling mechanism of GPR88, and will facilitate the innovative drug discovery for neuropsychiatric disorders and for deorphanization of this receptor.

[1] Kobilka Institute of Innovative Drug Discovery, School of Medicine, Chinese University of Hong Kong, Shenzhen, Guangdong 518172, China. [2] School of Life Sciences, University of Science and Technology of China, Hefei, Anhui 230026, China. [3] Department of Molecular and Cellular Physiology, Stanford University School of Medicine, Stanford, CA 94305, USA. [4] Graduate School of Pharmaceutical Sciences, Tohoku University, Sendai 980-8578, Japan. [5] Department of Chemistry and Pharmacy, Medicinal Chemistry, Friedrich-Alexander University Erlangen-Nürnberg, Nikolaus-Fiebiger-Straße 10, Erlangen 91058, Germany. [6] Warshel Institute for Computational Biology, School of Medicine, Chinese University of Hong Kong, Shenzhen, Guangdong 518172, China. [7] These authors contributed equally: Geng Chen, Jun Xu, Asuka Inoue, Maximilian F. Schmidt. ✉email: iaska@tohoku.ac.jp; peter.gmeiner@fau.de; liuzheng@cuhk.edu.cn; yangdu@cuhk.edu.cn

G-protein-coupled receptors (GPCRs) are the largest family of membrane signaling proteins in the human genome, with more than 800 members[1]. Approximately 140 of these receptors are orphan GPCRs (oGPCRs) whose endogenous ligands have not yet been identified[2,3]. Recent advances in structural biology have led to the determination of numerous high-resolution structures of GPCRs bound to antagonists or agonists, as well as complex structures with downstream signaling proteins, including G-protein and arrestin[4–6], which have significantly improved our understanding of GPCR ligand binding and activation mechanism at the molecular level. However, relatively little is known about the ligand recognition and signaling mechanism of oGPCRs due to the lack of tool ligands. Recent structural studies of the oGPCR GPR52 reveal a unique self-activation mechanism[7], suggesting that there might be some unknown mechanisms for ligand binding and signaling within these orphan receptors.

GPR88 is a brain-specific oGPCR of the class A rhodopsin family, with particular robust expression in the striatum[2,3,8]. GPR88 is able to modulate GABAergic and glutamatergic signaling and the activity of several other GPCRs such as dopamine receptors and opioid receptors[9,10]. Transcriptional profiling and knockout-mouse studies have shown that GPR88 plays important roles in regulating diverse brain and behavioral functions, such as cognition, mood, reward-based learning, and motor control[9,11,12]. Consequently, GPR88 is emerging as a potential drug target for the treatment of various human central nervous system (CNS) related diseases, including schizophrenia, Parkinson's disease (PD), bipolar disorder, anxiety, depression, and addiction[2,3,13].

GPR88 is distantly related to other well-studied class A GPCRs, with the highest similarity to 5-HT1D receptor (18% identity over the entire sequence)[2,8,10]. The predicted seven transmembrane segments for GPR88 are inconsistent among protein databases including UniProt and GPCRdb. Moreover, GPR88 lacks several features conserved in many other GPCRs such as the cysteines involved in the formation of disulfide bonds between the extracellular loops as well as the PIF motif[8]. These features indicate that GPR88 may be an atypical GPCR with a potentially different molecular mechanism for signal transduction. Despite extensive efforts in deorphanizing GPR88, its endogenous ligands remain unknown. Nevertheless, a family of synthetic agonists including (1R, 2R)-2-PCCA (hereafter denoted as 2-PCCA, Fig. 1a) and RTI-13951-33 was developed[13–17]. Cell signaling studies using these small molecular agonists indicated that GPR88 primarily couples to Gi/o proteins[15].

In an effort to understand the structural basis for GPR88 function and to provide a template for a structure-based design of novel leads and drug candidates, in this study, we determine the structures of the human GPR88-Gi1 signaling complex in the presence or absence of the synthetic agonist 2-PCCA using cryo-electron microscopy (cryo-EM). These structures, together with mutagenesis studies and an inactive state model, could provide a structural framework for understanding the ligand binding, activation, and signaling mechanism of GPR88.

## Results

**GPR88-Gi1 cryo-EM structure determination**. To improve cell surface expression, we fused a BRIL moiety to the N-terminus of the wild-type human GPR88 (Supplementary Fig. 1a). We initially utilized the agonist 2-PCCA[14] to stabilize the GPR88-Gi1 signaling complex (Fig. 1a). Using the NanoBiT-G-protein dissociation assay, we found that, among the four G-protein families, Gi1 was preferentially activated by GPR88 upon stimulation with 2-PCCA (Fig. 1b). The GTP turnover assay using purified proteins confirmed Gi1 activation by 2-PCCA in vitro (Fig. 1c).

Interestingly, we observed high-basal activity of GPR88 in the GTP turnover assay and the cell-based TGFα shedding assay (Fig. 1c and Supplementary Fig. 1b). We then assembled the GPR88-Gi1 complex with scFv16 (a Gi-stabilizing antibody) in the presence or absence of 2-PCCA, and obtained the cryo-EM density maps of the two complexes at a global nominal resolution of 2.4 Å and 3.0 Å, respectively (Supplementary Figs. 1c–h and 2a–f). These high-resolution maps allowed us to confidently build the atomic structures of the signaling complex (Fig. 1d, e, Supplementary Figs. 1i and 2g; Supplementary Table 1). The plotted snake diagram based on the transmembrane core regions shows that GPR88 has a long N and C terminus as well as a relatively long intracellular loop 3 (ICL3) (Supplementary Fig. 1a), while the densities of these regions were missing in our cryo-EM map. We also did not observe electron densities for ECL1 (extracellular loop 1) and most residues of the ECL2, indicating intrinsically flexible and disordered properties of these regions. The missing ECL2 density in GPR88 is in contrast to the recent structure of the orphan GPR52 in which the ECL2 region is well-folded and occupies the orthosteric pocket as a built-in agonist for self-activation[7]. Sequence alignment of ECL2 of GPR52 and GPR88 shows low homology (Supplementary Fig. 3a), suggesting that GPR88 may utilize a distinct self-activating mechanism. Interestingly, we observed an electron density in the canonical extracellular orthosteric site in both maps (Fig. 1d, e), which may confer the high-basal activity of GPR88. However, the density is not assigned in this study. To our surprise, we found that 2-PCCA binds to a herein identified pocket located in the membrane-facing surface of the cytoplasmic ends of TM5 and 6 (transmembrane segments 5 and 6) (Fig. 1e). Based on structural homology with other class A GPCR structures, we specify this unexpected binding site as an allosteric site hereinafter. The excellent electron density enabled unambiguous modeling for the 2-PCCA molecule in the allosteric binding pocket (Supplementary Fig. 3b). In addition, the map of the 2-PCCA-bound complex also reveals putative densities for three cholesterol molecules, which locate on the side of the 2-PCCA molecule (Supplementary Fig. 3c).

**Orthosteric and allosteric binding pockets of 2-PCCA**. The unassigned extracellular density located in a pocket created mainly by TM3, TM4, TM5, and TM7, as well as part of the ECL2 (Fig. 2a, b). As this position is generally recognized as the orthosteric binding site in class A GPCRs, we suspected that this density represents an endogenous ligand of GPR88 that was co-purified with the receptor. Electrostatic potential surface of the pocket shows that TM3, TM4, and TM5 create a long hydrophobic pore, while the extracellular surface is mainly charged and hydrophilic (Fig. 2c). Based on the shape of the density and the feature of the pocket, we suppose that the ligand is a lipid molecule with its non-polar tail inserts into the hydrophobic pore while the polar head group lies in the extracellular surface.

Previous structural studies have uncovered several allosteric binding sites on the surface of GPCRs[18–23]. Remarkably, 2-PCCA binds to a pocket formed by the cytoplasmic ends of TM5 and 6, as well as the extreme C-terminus of the α5 helix of Gi1 (Fig. 2d–g, Supplementary Fig. 3d), an allosteric binding site that has not yet been reported in other GPCRs. 2-PCCA is composed of a central amide that is substituted with three moieties: an aminoalkyl (**R1**), a pyridylcyclopropyl (**R2**), and a biaryl group (**R3**) (Fig. 1a). The majority of the contacts between 2-PCCA and GPR88 are mediated by hydrophobic interactions (Fig. 2f, g). The **R2** moiety of the allosteric 2-PCCA inserts into a pocket created by TM5, TM6 and the α5 helix of Gi1, and the ortho-nitrogen in the aromatic ring forms hydrogen bond interaction with the

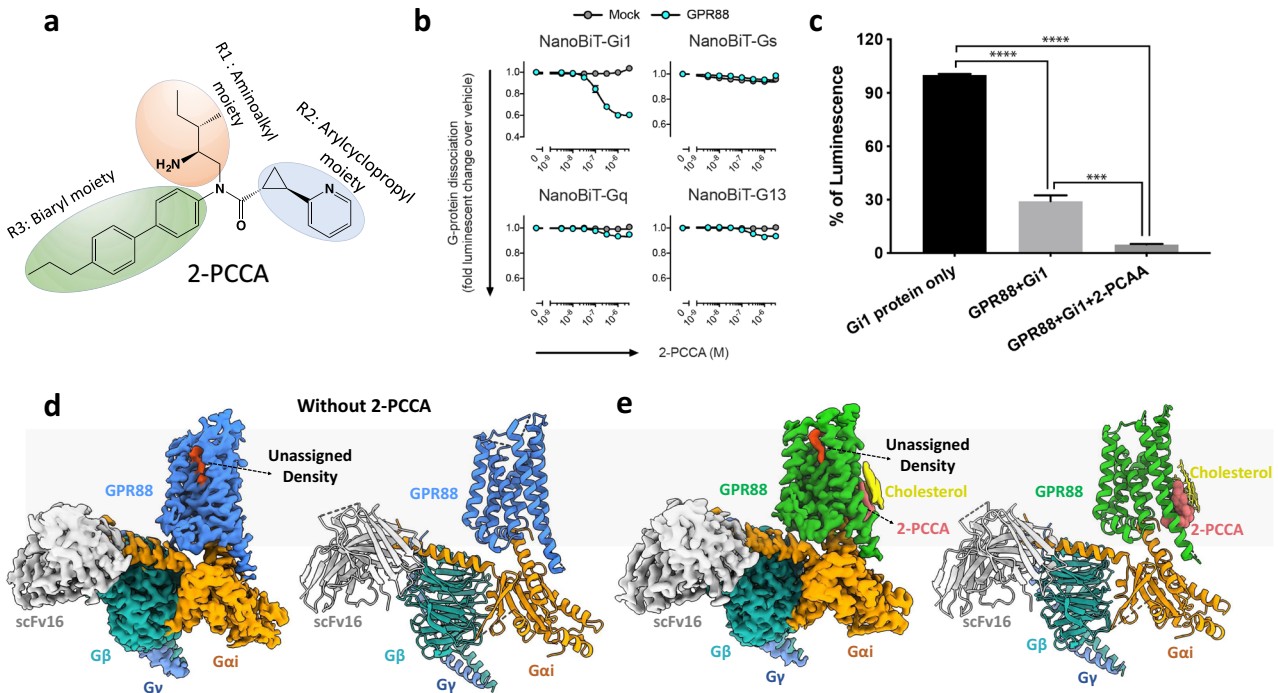

**Fig. 1 Cryo-EM structures of GPR88-Gi1 in the apo and 2-PCCA-bound forms. a** The chemical structure of 2-PCCA. **b** NanoBiT G-protein dissociation assay. Concentration–response curves of the G-protein dissociation signals for the indicated G-protein members. Symbols and error bars represent mean and s.e.m. of 3 (Gs), 4 (Gq, G13), and 6 (Gi1) independent experiments, each performed in duplicate. **c** Coupling of apo-state GPR88 or 2-PCCA bound GPR88 with Gi1 measured by GTPase Glo assay using purified proteins in detergent micelles. Error bars denote mean and s.e.m. of five independent experiments with repeats in triplicate. Statistical analyses were performed using the ordinary one-way ANOVA. ***$p < 0.001$; ****$p < 0.0001$. **d**, **e** Cryo-EM maps and structural models of GPR88-Gi1 signaling complex in the absence (**d**) or presence (**e**) of 2-PCCA.

backbone NH of G283[6.34]. The **R1** and **R3** moieties locate at the membrane-facing surface of TM5 and TM6, forming extensive hydrophobic contacts with the surrounding non-polar residues (V/I/L/C). In addition, the primary amine of **R1** forms a hydrogen bond with the backbone carbonyl of S282[6.33]. Of interest, three putative cholesterols are observed corresponding to the inner leaflet of the lipid bilayer. These cholesterols, together with the cytoplasmic ends of TM5 and TM6, create a similar hydrophobic pore as observed in the orthosteric pocket, which may further strengthen the binding of the allosteric 2-PCCA (Supplementary Fig. 3c).

The interactions between 2-PCCA and GPR88 observed in our structure correlate well with previous structure-activity relationship (SAR) studies with 2-PCCA derivatives[13]. For example, replacement of the primary amine of **R1** completely abolished the activity of the compound, which is consistent with the polar interactions with GPR88. Besides, the original lead compound bearing a phenyl substituent in **R2** displayed lower activity compared to the pyridine derivative. Replacement of the pyridine of **R2** with a cyclohexyl group can result in a complete loss of activity of the agonist. These data are in agreement with the hydrogen-bonding interaction between the pyridine nitrogen of 2-PCCA and G283[6.34] in the allosteric pocket. The SAR data also showed that moving the distal phenyl group of **R3** to the meta- or ortho-position of the internal benzene ring of **R3** can also lead to a significant loss of activity. Indeed, the shape of hydrophobic pore for allosteric pocket is most suitable for a para-substituted biaryl moiety (Fig. 2d–g). Moreover, replacement of the distal phenyl group of **R3** with polar substituents also significantly reduced the activity, which corresponds well to the fact that **R3** inserts into a hydrophobic pore. To further correlate the ligand activity with our structural observations, we introduced a number of mutations in the allosteric binding site and assessed their

effects on GPR88 function with the NanoBiT-G-protein dissociation assay (Fig. 2h–j, Supplementary Fig. 4). Mutations of L209[5.55], V216[5.62], V219[5.65], and L287[6.38] to alanine lead to significant loss of pEC$_{50}$ values, suggesting that these hydrophobic contacts in the allosteric pocket are crucial to GPR88 function. Notably, mutations of V216[5.62] into more bulky residues (F/L) result in a greater reduction of pEC$_{50}$ values than other mutations, and the G283V[6.34] mutation nearly abolishes the activity of 2-PCCA, which is likely due to a severe steric clash. The G283V[6.34] mutant displayed similar expression level and constitutive activity to those of the WT GPR88 (Fig. 2h, Supplementary Fig. 1b), indicating that the G283V[6.34] mutant may be fully responsive to a putative endogenous ligand and retains G-protein signaling activity.

As 2-PCCA possesses analogous chemical structure to lipid molecules, which has a polar head and a hydrophobic tail (Fig. 1a). We reasoned that 2-PCCA could compete with the putative endogenous ligand and bind to the orthosteric site as well. Of interest, 2-PCCA can be well docked into the orthosteric density, especially for the **R2** moiety, which fits well with the density in the extracellular surface (Supplementary Fig. 5a). Besides, an additional weak density was observed to fit the small **R1** moiety, and the long hydrophobic pore can well accommodate with the **R3** moiety (Supplementary Fig. 5a–c). To validate this possibility, we measured the activity of 2-PCCA on several mutants in the orthosteric pocket (Supplementary Figs. 4 and 5e). Most of the mutations in the extracellular surface dramatically reduced the cell-surface expression level of GPR88 (W84[2.56], G117[3.29], G121[3.33], W322[7.39]), suggesting that these residues are key for the functional expression of the receptor. One possibility is that binding of the putative endogenous ligand by these residues facilitates proper folding and/or sorting of GPR88 to the cell membrane. Although most of the mutations have little

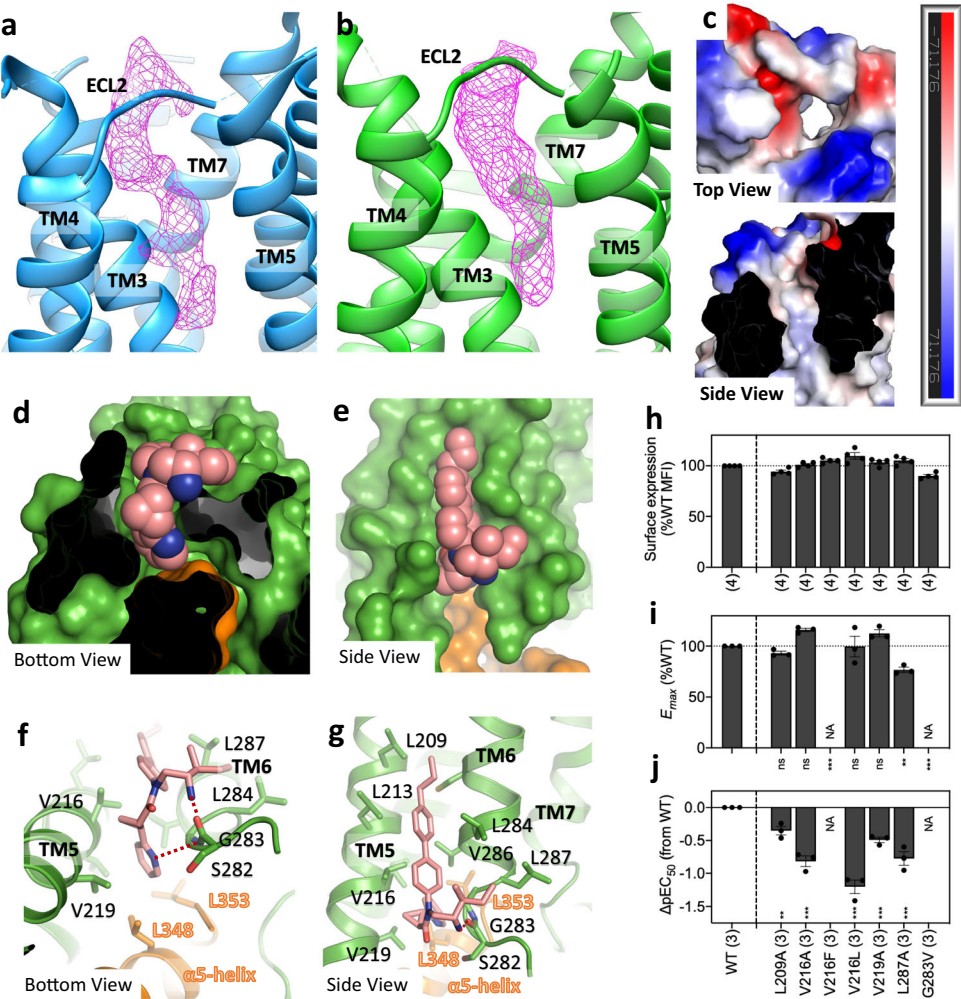

**Fig. 2 Orthosteric and allosteric binding sites. a, b** The unassigned electron density observed in the canonical orthosteric pocket of GPR88. Blue, structure without 2-PCCA (**a**); green, structure bound to 2-PCCA (**b**). **c** The charge distribution of the orthosteric pocket is shown in two different views. **d, e** The molecular surface of the allosteric pocket from bottom (**d**) and side (**e**) views. **f, g** Detailed interactions between 2-PCCA and the allosteric pocket from bottom (**f**) and side (**g**) views. Polar interactions are highlighted as dashed lines. **h–j** Cell-surface expression (**h**) and Gi-coupling activity (**i, j**) were analyzed by the flow cytometry and the NanoBiT-Gi dissociation assay, respectively. From the concentration–response curves (Supplementary Fig. 4), $E_{max}$ (**i**) and $\Delta pEC50$ (**j**) values relative to the wild type were calculated. Colors in the mutant bars indicate an expression level matching to that of titrated wild type. NA, parameter not available because of lack of the ligand response. Statistical analyses were performed using the ordinary one-way ANOVA followed by the two-sided Sidak's post hoc test with the expression-matched (colored) WT response. ns, $p > 0.05$; $*p < 0.05$; $**p < 0.01$; $***p < 0.001$. Bars and error bars represent mean and s.e.m. of 3 independent experiments, denoted as the parenthesis at the bottom of the figure panels. Source data are provided as a Source data file.

effect on 2-PCCA activity ($E_{max}$ or $\Delta pEC_{50}$), mutations of $W322^{7.39}$ and $W84^{2.56}$ to smaller alkyl residues (A/I/L/V) lead to loss of Gi activation by 2-PCCA, consistent with the model showing that the **R2** moiety forms aromatic stacking with these residues (Supplementary Fig. 5d, e). Besides, mutations of the residues $G117^{3.29}$ and $G121^{3.33}$ to bulky residues (F/L/W) displayed decreased $pEC_{50}$ of the ligand, consistent with the steric clash effect with the modeled 2-PCCA (Supplementary Fig. 5d, e). We also calculated binding free energy of the two equivalents of 2-PCCA with respect to their geometry center distances away from the experimental coordinates (Supplementary Fig. 5f). The calculated energy barrier was high when 2-PCCA only binds to either the orthosteric (route 2, 10.67 kcal/mol) or allosteric site (route 3, 5.83 kcal/mol). However, when both orthosteric and allosteric 2-PCCA approach the receptor in a coordinated manner via route 1, the barrier is reduced to 3.55 kcal/mol (note there are other possible routes for the coordinated binding). Together, these results suggest that the

orthosteric pocket could serve as a second binding site of 2-PCCA. Notably, a recent structure of the bile acid receptor (GPBAR) also revealed potential two-sites binding mode for the agonist INT-777. However, the molecule fitted in the electron density that located in a well-defined allosteric site formed by TM3, TM4, and TM5 remains uncertain and ambiguous due to the similarity among INT-777, cholesterol, and other bile acids[24].

**The active conformation of GPR88**. The overall structures of GPR88 with or without 2-PCCA are similar, with RMSD of 0.587 Å (Supplementary Fig. 6). As the 2-PCCA-bound structure has higher resolution, we used this structure for the following analysis. Structural comparison with active rhodopsin and other class A GPCRs shows that GPR88 has shorter transmembrane helices, most notable for TM6 (Supplementary Fig. 7a). The position of TM6 of GPR88 is more inward and the distance between TM5 and TM6 is larger than other class A GPCRs. As a consequence, a wide cavity is formed at the interface of the cytoplasmic ends of

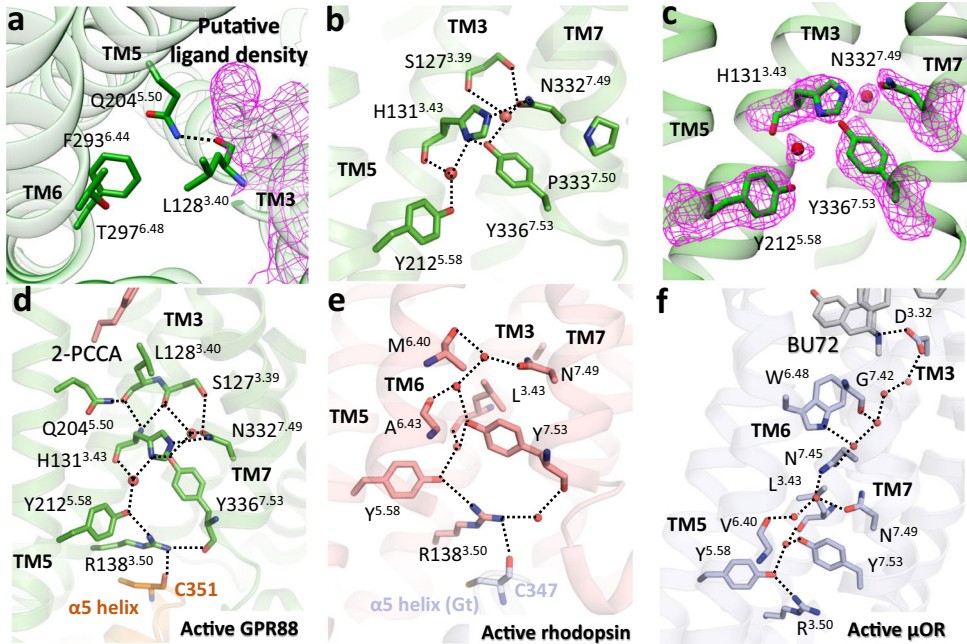

**Fig. 3 The active state GPR88 and the water-mediated hydrogen-bonding network. a** The active conformation of $T^{6.48}$ and the $Q^{5.50}$-$L^{3.40}$-$F^{6.44}$ triad. **b** The $N^{7.49}P^{7.50}xxY^{7.53}$ motif and two water molecules in the receptor core. **c** The electron densities of the two water molecules and surrounding residues. **d-f** Water-mediated hydrogen-bonding networks in the active GPR88 (**d**), rhodopsin (**e**, PDB: 2 × 72), and the µOR (**f**, PDB: 5C1M). Polar interactions are highlighted as dash lines.

TM5 and TM6, which allows the binding of the allosteric 2-PCCA (Supplementary Fig. 7b). Strikingly, this cavity is not seen in rhodopsin and other class A GPCRs (Supplementary Fig. 7b), indicating that the allosteric pocket is unique for GPR88 or certain GPCRs with similar structural features. In addition, sequence alignment shows that GPR88 not only lacks the cysteines in the extracellular loops but also lacks the rotamer toggle switch $W^{6.48}$ and the $P^{5.50}$-$I/L^{3.40}$-$F^{6.44}$ motif. Both entities are highly conserved in most rhodopsin family GPCRs (Supplementary Figs. 7c and 8a). The lack of cysteines involved in the disulfide bond formation likely makes the ECLs of GPR88 conformationally dynamic, explaining the missing densities in the cryo-EM map (Supplementary Fig. 1a). Previous structural studies have established a common activation mechanism of the rhodopsin family GPCRs in which the toggle switch $W^{6.48}$ triggers the outward movement of $TM6^{18,25,26}$, however, position 6.48 is a smaller threonine in GPR88. Besides, position 5.50 corresponds to a highly conserved proline in most class A GPCRs (Supplementary Fig. 8a), and it has been proposed that the rearrangement of the $P^{5.50}$-$I^{3.40}$-$F^{6.44}$ core triad plays a key role in the propagation of conformational changes from the extracellular domain to the G-protein coupling interface[27,28]. Interestingly, in GPR88, the conserved $P^{5.50}$ is replaced by a polar residue $Q204^{5.50}$ forming a hydrogen bond with the backbone carbonyl of $L128^{3.40}$ in the active conformation (Fig. 3a). Notably, $L128^{3.40}$ locates just below the unassigned density and likely has direct contacts with the putative endogenous ligand (Fig. 3a). Therefore, the $Q204^{5.50}$-$L^{3.40}$-$F^{6.44}$ triad in GPR88 may still play an important role in agonist-induced receptor activation (Fig. 3a). Indeed, mutation of $Q204^{5.50}$A leads to reduced signaling efficacy of GPR88 (Supplementary Fig. 8b–e). Of interest, $Q204^{5.50}$P is functional or even enhances the efficacy, indicating that a more typical $P^{5.50}$-$L^{3.40}$-$F^{6.44}$ motif may be more efficient for signal transduction than the $Q204^{5.50}$-$L^{3.40}$-$F^{6.44}$ motif (Supplementary Fig. 8b–e).

The $N^{7.49}P^{7.50}xxY^{7.53}$ motif in TM7 is another highly conserved sequence among class A GPCRs. In the high-resolution crystal

structure of the active rhodopsin and µ-opioid receptor (µOR), $Y^{7.53}$ and $N^{7.49}$ interact with $Y^{5.58}$ in TM5 and the backbone carbonyl of $L^{3.43}$ in TM3 via a water-mediated network[28,29]. Comparison of the side chains of $Y^{7.53}$, $N^{7.49}$, and $Y^{5.58}$ for most active GPCR structures suggests a similar polar network (Supplementary Fig. 7d). In the structure of GPR88, the conserved $L^{3.43}$ is replaced by a bulkier and hydrophilic $H131^{3.43}$, and the $Y336^{7.53}$ displays a rotamer distinct from other GPCRs and pointing towards TM3 to form a direct hydrogen bond with $H131^{3.43}$. In addition, the $N332^{7.49}$ forms a hydrogen bond with $H131^{3.43}$ and $S127^{3.39}$ in TM3 (Fig. 3b). Remarkably, we observed electron densities for two water molecules located on the top and bottom of $Y336^{7.53}$ and $H131^{3.43}$ interfaces, respectively (Fig. 3c). The top water further strengthens the polar interactions between the $N^{7.49}P^{7.50}xxY^{7.53}$ motif with TM3, while the bottom water mediates a similar hydrogen-bonding network among $Y336^{7.53}$, $H131^{3.43}$, and the conserved $Y212^{5.58}$ as observed in rhodopsin and the µOR structures (Supplementary Fig. 7d). While the $Y^{7.53}$ forms a hydrogen bond with the corresponding water in rhodopsin and µOR, $Y336^{7.53}$ in GPR88 participates in this hydrogen-bonding network through the sidechain of $H131^{3.43}$ (Fig. 3b). Moreover, $H131^{3.43}$ is linked to $Q204^{5.50}$ through a hydrogen bond network formed by the backbone amine of $H131^{3.43}$ and backbone carbonyl of $L128^{3.40}$ (Fig. 3d). These observations further suggest the role of this $Q^{5.50}$-$L^{3.40}$-$F^{6.44}$ motif in the signal transduction from the orthosteric pocket to the G-protein-coupling interface. Notably, mutation of $H131^{3.43}$ to L or I or A drastically reduced the cell surface expression of GPR88 (Supplementary Fig. 8c), probably due to the incompatibility between the hydrophobic alkyl chain and the polar network, suggesting distinctive structural characteristics of GPR88 from other class A GPCRs.

On the intracellular side of the $N^{7.49}P^{7.50}xxY^{7.53}$ motif, we found that $Y212^{5.58}$ forms a hydrogen bond with the conserved $R138^{3.50}$ in TM3, which further forms hydrogen-bonding interactions with the backbone carbonyl of $Y336^{7.53}$ and $C351^{G.H5.23}$

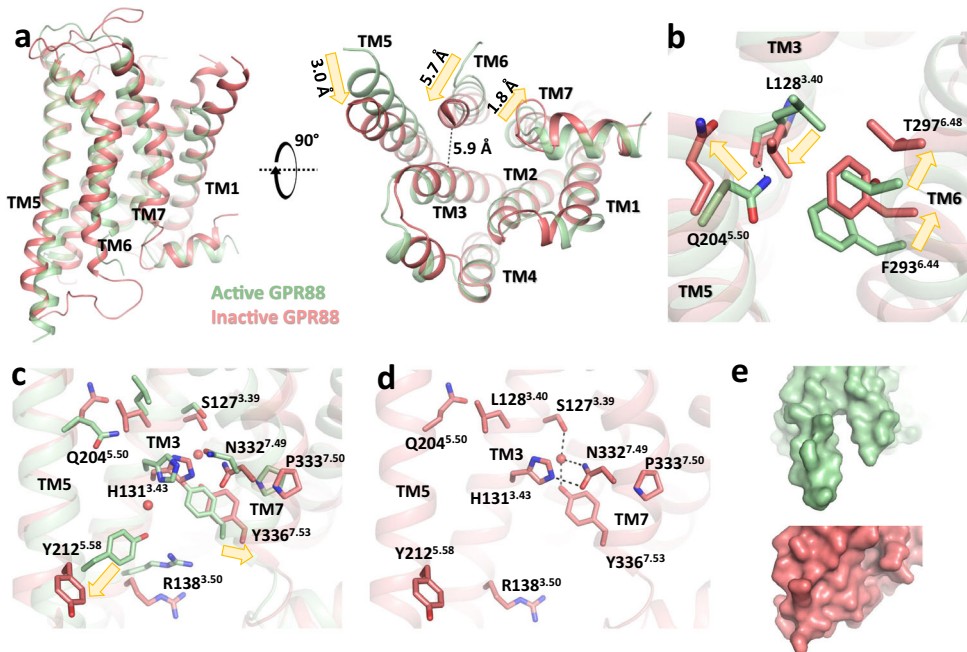

**Fig. 4 Metadynamics simulations of inactive-state GPR88. a** Comparison of overall structures of GPR88 in the inactive state (calculated, red) and the active state (cryo-EM, green). **b, c** Conformational changes of the Q5.50-L3.40-F6.44 triad (**b**) and the water-mediated hydrogen-bonding network (**c**) associated with receptor deactivation (yellow arrows), water molecules of the active cryo-EM shown as red spheres. **d** Polar networks of the inactive conformation (calculated). **e** Comparison of the surfaces of the allosteric pocket of inactive and active states. The PDB file of the calculated inactive GPR88 model is provided as Supplementary Data 1. Source data are provided as a Source data file.

(CGN numbering system) of Gi1 (Fig. 3d). This is similar to the polar network observed in the structure of constitutively active rhodopsin bound to the GαCT peptide (Fig. 3e)[29]. In the high-resolution crystal structure of μOR bound to the G-protein mimicking Nb35, the water-mediated polar network ends at R[3.50] (Fig. 3f), while the μOR-Gi1 complex structure shows that there is a lack of hydrogen-bonding interaction between R[3.50] and the backbone carbonyl of Y[7.53] and C351[G.H5.23] of Gi1[28]. In addition, the polar network in the μOR starts from the hydrogen bond between the orthosteric agonist BU72 and D[3.32] (Fig. 3f). However, the orthosteric pockets for GPR88 and rhodopsin are highly hydrophobic to accommodate the long lipophilic chain of 2-PCCA and retinal, respectively. The polar interaction between 2-PCCA and GPR88 is not connected to the hydrogen-bonding network (Fig. 3c, d), suggesting that agonist binding may be less important for the formation of the active polar network in GPR88 or rhodopsin than other typical GPCRs. Indeed, both GPR88 and rhodopsin have relatively high-basal activity[10,29] (Fig. 1c). Unlike the TM6 of rhodopsin and μOR, which heavily participate in the water-mediated polar network, the TM6 of GPR88 is devoid of this polar network in our structure (Fig. 3d–f). Perhaps there are additional unobserved water molecules in GPR88, and it is also possible that GPR88 may signal through a distinctive molecular mechanism, consistent with the different micro-switches observed in GPR88.

**Metadynamics simulations of inactive-state GPR88.** To further understand the conformational changes associated with GPR88 activation, we sought to obtain a model for the inactive-state of GPR88 using metadynamics simulations. This method is an attractive alternative to long-term unbiased MD simulations to investigate on conformational changes of GPCRs[28,30–32], as it allows enhanced sampling of rare events by accelerating conformational transitions and enables estimation of the free energy landscape of complex molecular systems[33]. To validate the

reliability of our simulation protocol, we first calculated the inactive states of three prototypical class A GPCRs, the β2 adrenergic receptor (β2AR), the M2 muscarinic receptor (M2R), and the μ-opioid receptor (μOR)[34–36]. According to the free energy landscape for all receptors, the energetically most favorable receptor conformation is found at a low TM3-TM6 distance, referring to an inward shifted TM6 (Supplementary Fig. 9a–d). For the reference receptors, large conformational changes occurred in the G-protein coupling domain and the receptor core resulting in inactive-like global minimum structures (Supplementary Fig. 9e). In fact, the models of β2AR, M2R, and μOR are very similar to the corresponding inactive X-ray crystal structures[37–39] (RMSD = 1.9–2.2 Å for transmembrane regions) and their key motifs (TM6: RMSD = 1.0–1.7 Å; N[7.49]P[7.50]xxY[7.53]: RMSD = 1.0–1.4 Å and P[5.50]-I[3.40]-F[6.44]/P[5.50]-V[3.40]-F[6.44]: RMSD = 0.6–1.3 Å) clearly indicated inactive-state properties (Supplementary Fig. 10). Notably, a local minimum is observable around 4 Å in all simulation systems (Supplementary Fig. 9a–d). The receptor conformation in this minimum resembles the global minimum model except that the TM3-TM6 distance is shorter (caused by a slight inward shift of TM3 towards TM6). The meaning of this receptor conformation is not clear, but it might represent an alternative inactive conformation. The existence of alternative inactive conformations have also been suggested in previous studies that applied MD simulations but also by NMR and DEER spectroscopy experiments[27,40–42]. Having evidence that the method is able to derive an inactive receptor structure starting from the active-state cryo-EM coordinates, we performed 8.64 μs of metadynamics simulations on GPR88 upon removing 2-PCCA, Gi1, and scFv16 from the complex and obtained a free energy landscape along with the TM3-TM6 distance with a global minimum at 5.9 Å (Supplementary Fig. 9a). Comparison with the active cryo-EM structure shows that large conformational changes occurred at TM5, TM6, and TM7 upon receptor deactivation (Fig. 4a). Upon inactivation, we observe an extension of the α-helical structure of TM6 by four amino acids from the C-terminal end of ICL3. Hence, the formal TM6 distance change is only 5.7 Å,

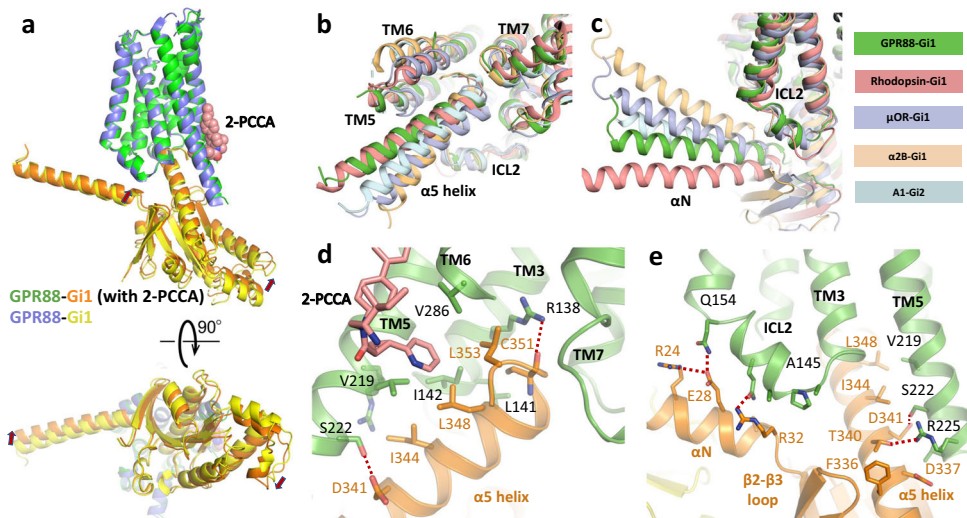

**Fig. 5 GPR88-Gi1 interface. a** Comparison of the GPR88-Gi1 complex with or without 2-PCCA. Red arrows indicate the shift of G-protein. **b**, **c** Superimposition of the receptor G-protein coupling interfaces for GPR88-Gi1, rhodopsin-Gi1 (PDB:6CMO), μOR-Gi1 (PDB:6DDE), α$_{2B}$AR-Gi1 (PDB:6K42), and A1R-Gi2 (PDB:6D9H). **d** Interactions of GPR88 with the α5 helix of Gαi1. **e** Rotated view of interactions of GPR88 with the α5 and αN helices of Gαi1. Polar interactions are highlighted as dash lines.

which is relatively small compared to other Gi-coupled receptors. We observed a 2.2 Å displacement of T297$^{6.48}$ and a large structural rearrangement of the Q$^{5.50}$-L$^{3.40}$-F$^{6.44}$ motif, suggesting that this region takes over the function of the P$^{5.50}$-L$^{3.40}$-F$^{6.44}$ core triad and, hence, is essential for GPR88 activation (Fig. 4b). Moreover, the water-mediated hydrogen bond network was rearranged in the global minimum. The water molecule at the bottom of H131$^{3.43}$ was displaced from the receptor core along with the conformational changes of H131$^{3.43}$, R138$^{3.50}$, Y212$^{5.58}$, and the N$^{7.49}$P$^{7.50}$xxY$^{7.53}$ motif while the top water molecule was still present, mediating a small polar network between TM3 and TM7 (Fig. 4c, d). Of note, H131$^{3.43}$ is also involved in this polar network in the inactive state model, suggesting its potential role in stabilizing the inactive conformation of GPR88. Notably, receptor deactivation reshapes the allosteric site (Fig. 4e). As a consequence, the hydrophobic pocket for the allosteric binding site is diminished in the simulated inactive-state model by the inward movement of TM6 (Fig. 4e), hence, preventing binding of 2-PCCA.

**Interaction of GPR88 and Gi1.** The complex structures of GPR88-Gi1 with or without 2-PCCA show almost the same G-protein coupling interface, with slightly upward shift of the G-protein in the 2-PCCA-bound structure (Fig. 5a). We used the 2-PCCA structure for the following analsysis of the interface. Recent cryo-EM structures of GPCR-Gi/o complexes suggest a diversity of the G-protein-coupling orientations. Indeed, the GPR88-Gi1 complex shows a distinct Gi1 orientation from other complexes (Fig. 5b, c). The overall interface of GPR88 and Gi1 consists of TM3, TM5-7, and ICL2 of GPR88, as well as the α5 and αN helices of the Gα subunit (Fig. 5b–e). Similar to previously reported complex structures, the C-terminus of the α5 helix inserts into the cavity formed by the cytoplasmic ends of TM3 and TM5-7. Hydrophobic residues I344$^{G.H5.16}$, L348$^{G.H5.20}$, and L353$^{G.H5.25}$, as well as C351$^{G.H5.23}$ on the wavy hook of α5 helix, interact with the TM3, TM5, and TM6 primarily through hydrophobic contacts, and an additional hydrogen bond between the C351$^{G.H5.23}$ backbone carbonyl of Gi1 and R138$^{3.50}$ of GPR88 (Fig. 5d). Remarkably, the allosteric 2-PCCA also participates in the hydrophobic network by interacting with both the Gi1 α5 helix and TM5-TM6 of GPCR88, which may further stabilize the interface of the GPR88-Gi1 signaling complex (Fig. 5d) and lead

to the slight shift of Gi1 (Fig. 5a). Of note, the amino acids involved in the interaction with GPR88 are mostly conserved across G-protein subtypes, especially at the C-terminal helix that forms the allosteric pocket with GPR88 (Supplementary Fig. 11a). Therefore, it is likely that this allosteric binding pocket still exists when GPR88 couples to the other Gi/o family G-proteins. Our structure provides the evidence that an allosteric ligand is directly involved in the interaction interface between receptor and G-protein.

In addition to the hydrophobic contacts, polar interactions are observed between the cytoplasmic end of TM5 and the bottom of the α5 helix, where T340$^{G.H5.12}$ and D341$^{G.H5.13}$ form hydrogen bonds with S222$^{5.68}$ and R225$^{5.71}$ in TM5, respectively. Another polar interface is observed between ICL2 of GPR88 and the αN helix of Gi1, where the two glutamines in ICL2 (Q154 and Q150) form hydrogen-bonding interactions with the two charged residues E28$^{G.HN.52}$ and R32$^{G.hns1.03}$, respectively (Fig. 5e). These polar interactions may be crucial for the coupling between GPR88 and Gi1. Of note, the interaction between ICL2 and Gαi subunit for GPR88-Gi1 is different from what has been observed for other non-rhodopsin Gi-coupled class A GPCRs. In those structures, the residue 34.51 of ICL2 engages into a hydrophobic pocket formed by the α5 and αN helices and the β2-β3 loop of Gαi (Supplementary Fig. 11b–d), although the interactions are relatively weak for several receptors[32,43–45]. By contrast, the residue 34.51 within GPR88 is a small alanine and is positioned away from the hydrophobic pocket on Gαi, similar to that of the rhodopsin-Gi complex (Supplementary Fig. 11e, f). These divergent features in the G-protein-coupling interface, together with a set of non-conserved micro-switches in the transmembrane core, further suggest that GPR88 may utilize a different mechanism for signaling transduction.

**Discussion**

We have determined the structure of the GPR88-Gi1 signaling complex in the presence or absence of the synthetic agonist (2-PCCA). These structures reveal a similar electron density within the canonical orthosteric pocket of GPR88, which may represent a putative endogenous ligand of the receptor. We find that 2-PCCA is an allosteric agonist that binds to the herein identified allosteric site and directly involves in the interaction with G-

protein, which further stabilizes the signaling complex, contributing the high activity of GPCR88. Notably, the well fitting between 2-PCCA and the orthosteric electorn density, together with mutagenesis and computational data suggest that 2-PCCA could also bind to the orthosteric site, revealing a potentially unusual drug binding mode in GPCRs. The high-resolution map of 2-PCCA-bound GPR88-Gi1 complex led us to the initial assumption that this synthetic agonist unambiguously occupies both the orthosteric and the allosteric pocket. However, a clear ligand density still exists in the orthosteric pocket of the apo-GPR88-Gi cryo-EM map, obtained as a control. The comparison of two maps suggested that a putative endogenous ligand in the orthosteric pocket may be co-purified with GPR88 and present in both our structures. This could serve as a caveat to the modeling GPCR ligands into cryo-EM maps and interpretation of the densities in the cryo-EM maps. Moreover, the shape of the unassigned density and the property of the orthosteric pocket suggest that GPR88 is likely a receptor in response to certain bioactive lipids. Of interest, recent structure of the sphingosine-1-phosphate (S1P) receptor bound to its endogenous agonist S1P reveals a similar orthosteric pocket as GPR88[46], where a long penetrating tunnel is formed for the binding of S1P (Supplementary Fig. 11g, h). It is possible that the endogenous agonist of GPR88 is a lipid molecule with similar structure to S1P, and that this lipid ligand may be able to bind to the allosteric site of GPR88 to modulate the signaling. However, we can not rule out the possibility that the lipid molecule has a branched structure and current electron-density maps merely show part of the density caused by conformational flexibility.

Our structure also reveals an extensive water-mediated hydrogen-bond network linking the receptor extracellular domain and the G-protein, which is important for stabilizing the active conformation of GPR88. Of not, we found that the non-conserved $H131^{3.43}$ in the transmembrane core of GPR88 not only plays pivotal role in mediating the polar network, but also is key for maintaining the functional expression of GPR88, uncovering a unique structure feature within this orphan GPCR. Comparison of the active cryo-EM structure with a validated inactive-state model of GPR88 generated by metadynamics revealed key conformational changes associated with GPR88 activation. Together, our studies provide a structural basis for understanding the ligand binding, activation, and signal transduction of the orphan receptor GPR88. These findings will facilitate the de-orphanization of GPR88. Moreover, a structure-based design of both agonists and antagonists may lead to valuable drug candidates for CNS diseases.

## Methods

### Expression and purification of GPR88

. The wild-type human GPR88 coding sequence (and all other cDNAs in this study) was synthesized by GENERAL BIOL (Chuzhou, China) and was cloned into pFastbac1 vector (Gibco) with an N-terminal Flag tag and a C-terminal His tag. To increase protein expression, the BRIL was fused into the N-terminal of GPR88. This N-terminal fusion strategy is widely used in enhancing GPCR recombinant expression and the previous study suggested such fusion generally does not affect receptor activity[47,48], and therefore we could regard this N-BRIL fusion construct as a surrogate of wild type receptor. The construct was transformed into DH10Bac to obtain the recombinant bacmid. The recombinant baculovirus was prepared in Sf9 insect cells using the Bac-to-Bac system. Sf9 cells were grown at 27 °C to a density of $4 \times 10^6$ per ml and infected with virus at a ratio of 1:40. Cells were collected after 48 h and stored at −80 °C until use.

For GPR88 purification, frozen cell pellets were lysed in 10 mM HEPES pH 7.5, 1 mM EDTA, 4 mg/ml iodoacetamide, 2.5 μg/ml leupeptin, 0.16 mg/ml benzamidine. Cell membranes were collected by centrifugation and solubilized in 20 mM HEPES pH 7.5, 100 mM NaCl, 1% LMNG, 0.1% CHS, 10% glycerol, 4 mg/ml iodoacetamide, 2.5 μg/ml leupeptin, 0.16 mg/ml benzamidine. After centrifugation to remove the insoluble debris, the supernatant was supplemented with 2 mM $CaCl_2$ and loaded onto anti-FLAG M1 affinity resin. The resin was extensively washed and the detergent concentration was reduced to 0.01% LMNG during the wash steps. The protein was eluted with 20 mM HEPES pH 7.5, 100 mM

NaCl, 0.01% LMNG, 0.001% CHS, 200 μM flag peptide, 5 mM EDTA. The elution fractions were concentrated and loaded onto Superdex 200 increase 10/300 size exclusion column (GE) with a running buffer of 20 mM HEPES pH 7.5, 100 mM NaCl, 0.01% LMNG, and 0.001% CHS. The peak fractions were collected and concentrated, fast-frozen in liquid nitrogen, and stored at −80 °C until use.

### Expression and purification Gi1 heterotrimer and scFv16

. For Gi1 heterotrimer expression, human Gαi1 was cloned into pFastbac1 vector (Gibco), and N-terminal 6 × His-tagged WT human Gβ1 and no-tag Gγ2 were cloned into a pFastBac-Dual vector (Gibco). The baculoviruses were prepared in the same way as GPR88. Trichoplusia ni Hi5 insect cells were grown at 27 °C to a density of $2.5 \times 10^6$ per ml and infected with both Gαi and Gβγ viruses at a ratio of 1:40 and 1:400, respectively. Cells were collected after 48 h and stored at −80 °C.

For the purification of Gi1 heterotrimer, cells were lysed in 10 mM HEPES pH 7.5 supplemented with 10 μM GDP and 1 mM $MgCl_2$. Cell membranes were collected and solubilized in 1% sodium cholate and 0.05% DDM supplemented with 25 μM GDP and 1 mM $MgCl_2$. After solubilization, the supernatant was collected and loaded onto a Ni-NTA resin. The resin was extensively washed and the detergent was exchanged to 0.08% DDM during wash step. Gi1 heterotrimer was eluted with 20 mM HEPES pH 7.5, 100 mM NaCl, 0.08% DDM, 250 mM imidazole, 100 μM TCEP, 25 μM GDP and 1 mM $MgCl_2$. After elution, 1 μL lambda phosphatase (NEB), 1 μL CIP (NEB), and 1 mM $MnCl_2$ was added and the mixture was incubated on ice overnight. The protein was then concentrated to ~20 mg/ml, fast-frozen in liquid nitrogen, and stored at −80 °C.

The scFv16 was purified as a secreted protein. The scFv16 sequence was cloned into pFastbac1 vector (Gibco) with an N-terminal GP67 secretion signal peptide and a C-terminal 8xHis tag. The baculovirus was prepared in the same way as for GPR88. Trichoplusia ni Hi5 insect cells were grown to a density of $2.5 \times 10^6$ per ml and infected with virus at a ratio of 1:40. After 60 h, the supernatant was collected and loaded onto a Ni-NTA resin. The resin was washed with 20 mM HEPES pH 7.5, 500 mM NaCl, and the protein was eluted by 20 mM HEPES pH 7.5, 500 mM NaCl and 250 mM imidazole. Elute protein was concentrated and loaded onto Superdex 200 increase 10/300 size exclusion column (GE). The peak fractions were collected and concentrated, fast-frozen in liquid nitrogen, and stored at −80 °C.

### GPR88-Gi1-scFv16 complex formation and purification

. For 2-PCCA-bound complex, 0.4 mg purified GPR88 was incubated with 1 mg Gi1 in a buffer composed of 20 mM HEPES pH 7.5, 100 mM NaCl, 1% LMNG, 100 μM agonist (1R,2R)-2-PCCA (MCE HY-100013A1) on ice for 2 h, then adding apyrase and 10 mM $MgCl_2$ and incubated on ice overnight to remove GDP. The next day, the mixture was diluted in a buffer of 20 mM HEPES pH 7.5, 100 mM NaCl, 0.01% LMNG, 0.003% GDN, 0.001% CHS, 10 μM (1R,2R)-2-PCCA, 2 mM $CaCl_2$ and loaded onto anti-FLAG M1 affinity resin. The resin was extensively washed and the detergent concentration was decreased to 0.003% LMNG with 0.001% GDN during the wash step. The complex was eluted with the 20 mM HEPES pH 7.5, 100 mM NaCl, 0.003% LMNG, 0.001% GDN, 0.004% CHS, 10 μM (1R,2R)-2-PCCA, 200 μM flag peptide, 5 mM EDTA and then incubated with 0.25 mg purified scFv16 for 2 h on ice. The GPR88-Gi1-scFv16 complex was finally loaded onto Superdex 200 Increase 10/300 size exclusion column (GE) against the running buffer composed of 20 mM HEPES pH 7.5, 100 mM NaCl, 0.003% LMNG, 0.001% GDN, 0.004% CHS, 10 μM (1 R,2 R)-2-PCCA, and 100 μM TCEP. The monomeric complex peak was collected and concentrated to 3 mg/ml for electron microscopy experiments. The complex without 2-PCCA was prepared in a same way without adding 2-PCCA in all steps. For the apo GPR88-Gi1 complex, it is exactly the same without adding 2-PCCA during complex assembly and the following purification steps.

### GTPase GLO assay

. For the GTPase-Glo assay, GPR88 was expressed and purified as described above and stored at −80 °C until use. The GTPase reaction was initiated by mixing Gi1 and GPR88 in 5 μL reaction buffer (20 mM HEPES, 100 mM NaCl, 0.02% LMNG, 1 mM $MgCl_2$, 5 μM GTP, 5 μM GDP, with or without 100 μM 2-PCCA in a 384-well plate. The final concentration of Gi1 was 0.5 μM and GPR88 was 4 μM, respectively, in the reaction system. For every independent experiment, Gi1 alone was set as a reference. The GTPase reaction was incubated at room temperature (22–25 °C) for 2 h. After incubation, 5 μL reconstituted 1xGTPase-Glo reagent (Promega) was added to the completed GTPase reaction, mixed briefly and incubated with shaking for 30 min at room temperature (22–25 °C) to convert the remaining GTP into ATP. Then 10 μL detection reagent (Promega) was added to the system and incubated in the 384-well plate for 5–10 min at room temperature (22–25 °C) to convert the ATP into luminescent signals. Luminescence intensity was quantified using a Multimode Plate Reader (PerkinElmer EnVision 2105) luminescence counter. Data were analyzed using GraphPad Prism 7.0.

### Cryo-EM sample preparation and data collection

. The amorphous alloy film[49] (CryoMatrix nickel titanium alloy film, R1.2/1.3, Zhenjiang Lehua Electronic Technology Co., Ltd.) was glow discharged at Tergeo-EM plasma cleaner. 3 μL purified complex sample was applied onto the grid and then blotted for 3 s with blotting force of 0 and quickly plunged into liquid ethane cooled by liquid nitrogen

using Vitrobot Mark IV (Thermo Fisher Scientific, USA). Cryo-EM data were collected at the Kobilka Cryo-EM Center of the Chinese University of Hong Kong (Shenzhen), on a 300 kV Titan Krios Gi3 microscope. The raw movies were recorded by a Gatan K3 BioQuantum Camera at the magnification of 105,000, The pixel size is 0.83 Å. Inelastically scattered electrons were excluded by a GIF Quantum energy filter (Gatan, USA) using a slit width of 20 eV. The movie stacks were acquired with the defocus range of −1.0 to −2.0 micron with a total exposure time 2.5 s fragmented into 50 frames (0.05 s/frame) and with the dose rate of 21.2 e/pixel/s. The semi-automatic data acquisition was performed using SerialEM[50].

**Image processing and model building**. For complex bound to 2-PCCA, the general strategy in the image processing follows the method in a hierarchical way as described[51,52]. Data binned by 4 times is used for micrograph screening and particle picking. The data with 2-time binning is used for particle screening and classification. The particle after initial cleaning was subjected to extraction from the original clean micrograph and the resultant dataset was used for final cleaning and reconstruction. Raw movie frames were aligned with MotionCor2[53] using a $9 \times 7$ patch and the contrast transfer function (CTF) parameters were estimated using Gctf and ctf in JSPR[54]. Only the micrographs with consistent CTF values including defocus and astigmatism were kept for following image processing. This process kept 5778 micrographs from 6215 raw movies. Templates for particle selection were generated by projecting the 3D volume of the AVP-V2R-Gs complex[55]. The 4,647,118 particles picked from template picking were subjected to 2 rounds of 2D classification, reducing their size to 1,706,690, and then reducing to 1,333,021 by 3D-classification. After several rounds of ab initio refinement, the particles kept to 988,958 were subjected to non-uniform refinement for a 2.44 Å reconstruction. The image parameters were converted back and to Relion[56] and cryoSPARC[57] by use of the pyem package.

For complex without 2-PCCA, a total of 3539 image stacks were collected were subjected to patch motion correction and patch CTF refinement. 3511 micrographs were selected for subsequence data processing. 3,166,931 particles were auto-picked and then subjected to 2D classification followed by ab initio reconstruction and heterogeneous refinement. The resulting 326,087 particles were subject to non-uniform refinement and yielded a map at 3.19 Å. Extracting with larger paticle box size results in 314,834 particles, which were subjected to non-uniform refinement and yielded a map at 2.98 Å.

The initial model of active-state GPR88 was built by SWISS-MODEL. The coordinates of Gi1 and scFv16 from μOR (PDB ID 6DDE) were used as templates. All models were docked into the EM density map using UCSF Chimera version 1.12, followed by iterative manual building in Coot[58] and refinement in Phenix[59]. The final model statistics were validated by Molprobity[60].

**NanoBiT G-protein dissociation assay**. GPR88-induced G-protein dissociation was measured by a NanoBiT-G-protein dissociation assay[61], in which the interaction between a Gα subunit and a Gβγ subunit was monitored by the NanoBiT system (Promega). Specifically, a NanoBiT-Gi1 protein consisting of Gαi1 subunit fused with a large fragment (LgBiT) at the α-helical domain (between the residues 91 and 92 of Gαi1) and an N-terminally small fragment (SmBiT)-fused Gγ2 subunit with a C68S mutation was expressed along with untagged Gβ1 subunit and GPR88. HEK293A cells were seeded in a 6-well culture plate at a concentration of $2 \times 10^5$ cells ml⁻¹ (2 ml per well in DMEM (Nissui) supplemented with 10% fetal bovine serum (Gibco), glutamine, penicillin, and streptomycin), 1 day before transfection. Transfection solution was prepared by combining 5 μL (per dish hereafter) of polyethylenimine (PEI) Max solution (1 mg ml⁻¹; Polysciences), 200 μL of Opti-MEM (Thermo Fisher Scientific) and a plasmid mixture consisting of 200 ng GPR88 (or an empty plasmid for mock transfection), 100 ng LgBiT-containing Gαi1 subunit, 500 ng Gβ1 subunit, and 500 ng SmBiT-fused Gγ2 subunit (C68S). After incubation for 1 day, the transfected cells were harvested with 0.5 mM EDTA-containing Dulbecco's PBS, centrifuged, and suspended in 2 ml of HBSS containing 0.01% bovine serum albumin (BSA; fatty acid-free grade; SERVA) and 5 mM HEPES (pH 7.4) (assay buffer). The cell suspension was dispensed in a white 96-well plate at a volume of 80 μL per well and loaded with 20 μL of 50 μM coelenterazine (Carbosynth) diluted in the assay buffer. After a 2 h incubation at room temperature, the plate was measured for baseline luminescence (SpectraMax L with SoftMax Pro 7.0.3 software, Molecular DeMvices) and titrated concentrations of (1R,2R)-2-PCCA (20 μL; 6X of final concentrations) were manually added. The plate was immediately read at room temperature for the following 5 min as a kinetics mode, at measurement intervals of 20 s. The luminescence counts from 3 to 5 min after ligand addition were averaged and normalized to the initial count. The fold-change values were further normalized to those of vehicle-treated samples and used to plot the G-protein dissociation response. Using the Prism 8 software (GraphPad Prism), the G-protein dissociation signals were fitted to a four-parameter sigmoidal concentration–response curve with a constrain of the Hill-Slope to absolute values <1.5. For each replicate experiment, the parameters Span (=Top – Bottom) and pEC₅₀ (negative logarithmic values of EC₅₀ values) of individual GPR88 mutants were normalized to those of WT GPR88 performed in parallel and the resulting $E_{max}$ values and the $\Delta pEC_{50}$ values were used to calculate ligand response activity of the mutants.

**Flow cytometry analysis**. Transfection was performed according to the same procedure as described in the "NanoBiT-G-protein dissociation assay" section. One day after transfection, the cells were collected by adding 200 μl of 0.53 mM EDTA-containing Dulbecco's PBS (D-PBS), followed by 200 μl of 5 mM HEPES (pH 7.4)-containing Hank's balanced salt solution (HBSS). The cell suspension was transferred to a 96-well V-bottom plate in duplicate and fluorescently labeled with an anti-FLAG epitope (DYKDDDDK) tag monoclonal antibody (Clone 1E6, FujiFilm Wako Pure Chemicals; 10 μg ml⁻¹ diluted in 2% goat serum- and 2 mM EDTA-containing D-PBS (blocking buffer)) and a goat anti-mouse IgG secondary antibody conjugated with Alexa Fluor 488 (Thermo Fisher Scientific, 10 μg ml⁻¹ diluted in the blocking buffer). After washing with D-PBS, the cells were resuspended in 200 μl of 2 mM EDTA-containing-D-PBS and filtered through a 40-μm filter. The fluorescent intensity of single cells was quantified by an EC800 flow cytometer (Sony). The fluorescent signal derived from Alexa Fluor 488 was recorded in an FL1 channel, and the flow cytometry data were analyzed with the FlowJo software (FlowJo). Live cells were gated with a forward scatter (FS-Peak-Lin) cut-off at the 390 setting, with a gain value of 1.7. Values of mean fluorescence intensity (MFI) from ~20,000 cells per sample were used for analysis. For each replicate experiment, MFI counts of GPR88 mutant samples were normalized to those of WT GPR88 (100% level) and the mock-transfected samples (0% level), and the resulting values were used to denote surface expression levels of the mutants.

**TGFα shedding assay**. To measure the constitutive activity of GPCRs, we used the TGFα shedding assay, which measures accumulation of Gq/11 and G12/13 signaling, as described previously[62,63]. To detect Gi-coupled GPCR, we utilized a chimeric Gαq/i1 subunit consisting of the Gαq backbone and the Gαi1-derived 6 amino acids at the C-terminus, which is capable of binding to Gi-coupled GPCRs and induces Gq signaling. Briefly, HEK293 cells were seeded in a 96-well cell culture plate at a concentration of $4 \times 10^5$ cells per ml in Opti-MEM I Reduced Serum Media (Thermo Fisher Scientific), in a volume of 80 μl per well. A transfection mixture was prepared by mixing the PEI transfection reagent (0.2 μl per well) and plasmids (20 ng alkaline phosphatase-tagged TGFα (AP-TGFα) plasmid, titrated GPCR plasmid (0.5 to 8 ng), and an empty pcDNA3.1 plasmid to balance the total plasmid volume, with or without 4 ng of the chimeric Gαq/i1 subunit) in Opti-MEM I Reduced Serum Media (20 μl). The transfection solution was added to the cells. For each condition, we used 4 replicate wells. After incubation for 1 day, the cell plate was spun at $190 \times g$ for 2 min and the conditioned media (80 μl per well) were transferred to an empty 96-well plate (conditioned media (CM) plate). The AP reaction solution (10 mM p-nitrophenylphosphate (p-NPP), 120 mM Tris–HCl (pH 9.5), 40 mM NaCl, and 10 mM MgCl₂) was dispensed into the cell plates and the CM plates (80 μl per well). The absorbance at 405 nm (Abs₄₀₅) of the cell plate and the CM plate was measured, using a microplate reader (SpectraMax 340 PC384, Molecular Devices), before and after a 40 min incubation at room temperature. AP-TGFα release was calculated as described previously[63] and the signal in the mock-transfected conditions was set at the baseline. As a positive control for spontaneous Gi-coupled GPCR, we used a M4-DREADD (M4D), which loses affinity to the endogenous ligand acetylcholine, and introduced glutamine mutant at L123³·⁴³, which is known to cause constitutive activity in other GPCRs[62].

**Metadynamics simulations**. Co-crystallized ligands and the intracellularly binding proteins were removed from the active-state Cryo-EM structures of β2AR (PDB: 6NI3)[34], M2R (PDB: 6OIK)[35], μOR (PDB: 6DDE)[36], and GPR88 (reported in this work). Missing loops were modeled using MODELLER software[64]. The long and flexible intracellular loop 3 (ICL3) was modeled as an oligopeptide of alternating glycines and serines. For the μOR, the natural ICL3 sequence was modeled since it shows a rather short ICL3 containing only 5 amino acids. For GPR88, modeling of the relatively large gap in ECL1 was omitted but with Q87 to L91 modeled to the extracellular tip of TM2. The residues R31-L36 were modeled at the extracellular tip of TM1.

All open ends of the amino acid sequence were end-capped with an acetyl- or N-methyl group at the terminal amines or carboxylic acids, respectively. All titratable residues were left in their dominant protonation state at pH 7.0 with the exception of E122³·⁴¹ of the β2AR, since it is located within the phospholipid bilayer facing a hydrophobic environment and thus likely to be protonated. Since a sodium ion within an allosteric binding site around D²·⁵⁰ is proposed to stabilize the inactive conformation of GPCRs, a sodium ion was modeled into this cavity[65–67]. This was achieved by transferring the coordinates of the sodium ion after alignment with the adenosine A₂A receptor inactive-state X-ray crystal structure (PDB-ID: 5IU4)[68]. This structure was chosen because it has the highest resolution (1.72 Å) of all published inactive GPCR structures according to GPCRdb, in which a sodium ion could be resolved in the cavity around D²·⁵⁰.

Parameter topology and coordinate files were generated using the tleap module of the AMBER18 program package[69]. The created GPCR models were energy minimized using the PMEMD module of AMBER18 by applying 500 steps of steepest decent followed by 4500 steps of conjugate gradient and subsequently converted to GROMACS input files. The GPCR models were aligned to their respective orientation of proteins in membranes (OPM)[70] structure (GPR88 was aligned to the OPM structure of the G-protein-bound β₂AR, PDB-ID: 3SN6[71]) and inserted into a

solvated and pre-equilibrated membrane of dioleyl-phosphatidylcholine (DOPC) lipids via the GROMACS tool g_membed[72]. Water molecules were replaced by sodium and chloride ions to result in neutral and physiological systems with 0.15 M NaCl. Final dimensions of the simulation systems were about $80 \times 80 \times 100$ Å containing ~65,200 atoms, including ~154 DOPC molecules, ~13,260 waters, ~58 sodium, and ~75 chloride ions. The prepared simulation systems were energy minimized and equilibrated using the NVT ensemble at 310 K for 1.0 ns followed by the NPT ensemble for 1.0 ns with harmonic restraints of 10.0 kcal·mol⁻¹ on the protein. In the NVT ensemble, the V-rescale thermostat was used. In the NPT ensemble the Berendsen barostat, a surface tension of 22 dyn·cm⁻¹, and a compressibility of $4.5 \times 10^{-5}$ bar⁻¹ was applied. The systems were further equilibrated for 25 ns with restraints on protein backbone atoms. Here, the restraints were reduced in a stepwise fashion to be 10.0, 5.0, 1.0, 0.5, and 0.1 kcal·mol⁻¹, respectively. To retain an active conformation of the G-protein interface, position restraints of 10.0 kcal·mol⁻¹ were applied to all receptor residues within 5 Å of the G-protein during equilibration.

The equilibrated GPR88 was further subjected to a 2.0 μs unbiased MD simulation to ensure a stable receptor model. The 10.0 kcal·mol⁻¹ restraints on the G-protein interface were maintained. A cluster analysis was applied to the trajectory by means of the CPPTRAJ module of AMBER18, omitting the first 500 ns. A representative frame of the main cluster was used for the following deactivation simulations.

To obtain an inactive model of the β2AR, M2R, μOR, and GPR88, a combined approach of single- and multiple-walker well-tempered (WT) metadynamics simulations was applied without any restraints[73,74]. The TM3-TM6 distance between the alpha carbons of $R^{3.50}$ and position 6.34 was used as Collective Variable (CV, reaction coordinate). Initially, multiple independent WT single-walker metadynamics simulations for each receptor with 50 ns each were performed. Gaussian hills with an initial height of 0.239 kcal·mol⁻¹ applied every 1.0 ps were used. The hill width was set to 1.0 Å. The Gaussian functions were rescaled in the WT scheme using a bias factor of 50. Using 32 frames (= walkers) extracted from these initial simulations for each receptor, WT multiple-walker metadynamics simulations were started. The walkers covered various receptor conformations ranging from an inward to outward shifted TM6. For the multiple-walker metadynamics simulations, the bias factor was reduced to 20. After a total simulation time of 10.56 μs for β2AR, M2R, and μOR and 8.64 μs for GPR88, the multiple-walker metadynamics simulations were stopped and the free energies were calculated using the sum_hills utility of the PLUMED plugin.

For all simulations, the lipid 14 force field[75] was used for DOPC molecules and ff14SB[76] for protein residues. The SPC/E water model was applied[77]. All simulations were conducted with GROMACS 2018.4 patched with PLUMED 2.5.0[78] using periodic boundary conditions and a time step of 2 fs with bonds involving hydrogen constrained using LINCS[79]. Long-range electrostatic interactions were computed using the particle mesh Ewald (PME)[80] method with interpolation of order 4 and fast Fourier transform (FFT) grid spacing of 1.6 Å. Non-bonded interactions were cut off at 12.0 Å.

**PDLD/S-LRA binding free energy calculations**. To evaluate the binding free energy of the two ligands with respect to their distances away from the experimental binding sites, we utilized the scaled semi-macroscopic Protein Dipole Langevin Dipole (PDLD) method[81,82], which is implemented in the MOLARIS-XG package[83,84]. The PDLD method can calculate binding free energies by constructing proper thermodynamic cycles. The energy is further evaluated by a linear response approximation (LRA), during which the energy is averaged over charged and uncharged configurations. In this work, we scaled the electrostatic energy with a dielectric constant of $\varepsilon = 4$ for the protein. We performed distances scans for both ligands during their dissociation from the binding sites. The two dissociation degrees of freedom are coupled together to generate the binding free energy surface. The electrostatic potential (ESP) charge distribution of ligand atoms is calculated by Gaussian with B3LYP/6-31G(d) method. We relaxed the systems for 0.1 ns using molecular dynamics before binding free energy evaluations.

**Reporting summary**. Further information on research design is available in the Nature Research Reporting Summary linked to this article.

## Data availability
The 3D Cryo-EM density maps of the 2-PCCA-bound and apo GPR88-Gi-scFV16 complex have been deposited in the Electron Microscopy Data Bank database under accession codes EMD-31164 and EMD-32904, respectively. The atomic coordinates for the atomic models of the 2-PCCA-GPR88-Gi-scFV16 complexes generated in this study have been deposited in the Protein Data Bank database under accession codes 7EJX and 7WZ4, respectively. The structural models of rhodopsin-Gi1, μOR-Gi1, α2BAR-Gi1, and A1R-Gi2 used in this study are available in the Protein Data Bank database under accession codes 6CMO, 6DDE, 6K42, and 6D9H, respectively. The PDB file of the calculated inactive GPR88 model is provided as Supplementary Data 1. Source data are provided with this paper.

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

## Acknowledgements

We would like to thank the Kobilka Cryo-Electron Microscopy Center, the Chinese University of Hong Kong, Shenzhen for our cryo-electron microscopy. Y.D. is supported by grants from Science, Technology and Innovation Commission of Shenzhen Municipality (Project code JCYJ20200109150019113 and SZBL2020090501011), and in part by the Kobilka Institute of Innovative Drug Discovery and Presidential Fellowship at the Chinese University of Hong Kong, Shenzhen. Z.L. is supported by the Kobilka Institute of Innovative Drug Discovery at the Chinese University of Hong Kong, Shenzhen. We thank Kouki Kawakami, Kayo Sato, Shigeko Nakano, and Ayumi Inoue (Tohoku University) for their assistance with plasmid preparation, maintenance of the cultured cells and the cell-based GPCR assays. We gratefully acknowledge the compute resources provided by the Erlangen Regional Computing Center (RRZE). A.I. was funded by the PRIME 19gm5910013, the LEAP JP19gm0010004, the BINDS JP20am0101095, and FOREST Program JPMJFR215T from the Japan Agency for Medical Research and Development (AMED); the Promotion of Science (JSPS) KAKENHI grants 21H04791, 21H05113, JPJSBP120213501, and JPJSBP120218801; Moonshot Research and Development Program JPMJMS2023; Daiichi Sankyo Foundation of Life Science; The Uehara

Memorial Foundation; Ono Medical Research Foundation; Takeda Science Foundation. M.F.S. and P.G. were supported by the DFG grant GRK 1910.

## Author contributions

G.C. designed the expression constructs, purified proteins, assembled the complex, performed GTPase Glo assay, made all mutations for the G-protein dissociation assay and TGFα shedding assay, analyzed the structure, and participated in method writing. J.X. and G.C. prepared cryo-EM grids, participated in grids screening and cryo-EM images collection, performed the model building and refinement, analyzed the structure, and prepared the figures. A.I. performed the G-protein dissociation assay, the TGFα shedding assay and the flow cytometry analysis, analyzed the data and participated in manuscript writing. M.F.S. developed the metadynamics-based deactivation protocol, calculated the inactive-state structure of GPR88 and participated in the preparation of figures under the supervision of P.G. C.B. performed free energy calculations of ligand binding. Q.L. participated in cryo-EM images collection. Z.L. supervised cryo-EM image collection, processed cryo-EM data, and participated in the preparation of supplementary figures. Y.D. provided overall project supervision. J.X. and Y.D. wrote the manuscript with input from all authors.

## Competing interests

The authors declare no competing interests.
