## [Peer Review File · Nature Communications]

Activation and allosteric regulation of the orphan GPR88-Gi1 signaling complexREVIEWER COMMENTS

Reviewer #1 (Remarks to the Author):

The manuscript entitled “Structure of the orphan GPR88-Gi complex reveals unique ligand binding and signaling mechanism” by Chen and colleagues is well written and clear. The authors describe the cryoEM structure of a complex including the orphan G protein coupled receptor (GPCR) GPR88, the Gi heterotrimer, and the recently discovered synthetic ligand (1R, 2R)-2-PCCA. Important features described in the present paper include the identification of 2 ligand binding pockets that intriguingly bind 2 molecules of 2-PCCA. Moreover, for the first time it is shown how the $\alpha 5$ helix in the G α i1 C-terminus contributes to one of the pockets highlighting a novel mechanism of G protein activation. The paper is of high importance as orphan receptors are untapped pharmacological targets with relevant physiological roles and this is especially true for GPR88. At the same time, the only orphan GPCR structure currently available is that of GPR52, thus new information related to other orphan GPCRs may guide deorphanization efforts leading to new insights into orphan receptor biology.

Comments:

The orthosteric binding site is defined as the site of interaction with endogenous ligands, while molecules that bind to the allosteric site normally only modulate signals initiated by agonists binding to the orthosteric site, without having any intrinsic signaling property. In the manuscript, this terminology is used as a proxy for extracellular binding pocket (orthosteric) or intramembrane binding pocket (allosteric). Considering that we do not know where the endogenous ligand binds in GPR88 (it could be one site, the other, or both), this terminology seems to be inappropriate especially in a case like the one shown here, where the same ligand binds to both sites. Experiments where point mutations in the proposed allosteric pocket (L2095.55, V2165.62, V2195.65 and L2876.38 to alanine) reduced the EC50 without affecting GPR88 membrane targeting and Emax (Fig. 2I-K, and S3A) suggest it could be an allosteric site. However, mutations V216F and G283V (that do not affect GPR88 membrane localization) completely eliminate Gi signaling supporting an active role of this “allosteric” site in the generation of GPR88 signal. The high constitutive activity shown for G283V does not represent an appropriate control as signal can be activated by intracellular locations.

Moreover, loss of Gi activation by 2-PCCA due to mutations in the “orthosteric” site (for example: W3227.39 and W842.56 to smaller alkyl residues) are inconclusive, as a >90% reduction of GPR88 is observed at the plasma membrane (it seems like there is a strong correlation between GPR88 membrane targeting and Emax in Fig. 2I-J). Competition assays using a different agonist that do not bind to both pockets would help to establish which one is the actual orthosteric site. If sites are named because of structural homology with other GPCRs, this should be made clear in the manuscript.

Statistical analysis applied to the data shown in Fig. 1C, Fig 2I-K, Fig S1B, and Fig. S5C-E is not reported anywhere in the manuscript.

In the discussion it is speculated that a lipid may be the endogenous ligand of GPR88. 2-PCCA interacts with GPR88 in both the identified pockets. Based on these information, could it be speculated that the 2 binding sites bind an endogenous ligand that interacts with both orthosteric and allosteric pockets simultaneously? Possibly a dimer?

The observation that the allosteric pocket is formed by GPR88 and Gi1 raises a question about the possible contribution of other G alpha proteins (Gi2, Gi3, Go, and Gz) in the formation of this binding pocket. Are the amino acids involved in the interaction conserved among other G proteins? Or this configuration is possible only with Gi1 creating an intrinsic bias in GPR88 activation towards this G protein?

Minor comments:

Line 91-93: Please consider including references to the studies demonstrating Gi/o coupling of GPR88.

Line 103: Reference to Fig. 1A is missing.

Lines 121-126: It is not clear why GPR88 and GPR52 should have similarities in the ECL2 structure and function, besides the fact that they are both class A GPCRs and orphans. Is there a high homology of amino acid sequence in this region between GPR88 and GPR52? A clarification would be helpful.

Line 213: mutant P177G is reported as P178G in Figure 2I and Supplementary Figure S3A.

Line 327: Please consider including references supporting this statement.

Line 376-378: This is not completely true, as mutation H131A shows a drastic reduction in membrane targeting (>60%), but similar, if not better, Emax compared to WT (Fig. S5B-E).

Methods: The origin of the DNA constructs used in the study (GPR88, Gi1, scFv16, etc.) should be reported.

Reviewer #2 (Remarks to the Author):

In this manuscript, cryo-EM structure was determined for orphan GPR88 in complex with a synthetic agonist 2-PCCR and the Gi1 protein. Two ligand molecules bound simultaneously to the orthosteric site and a newly identified allosteric binding site formed by the TM5 and TM6 intracellular domains and the

C-terminus of the Gi1 $\alpha 5$ helix. Notably, metadynamics simulations were performed to predict an inactive state model of GPR88.

Overall, the structure is very interesting and the manuscript is well written. However, a number of major suggestions that may help improving the manuscript include:

1. The authors mentioned that GPR88 exhibits high basal activity, suggesting that the apo receptor likely samples an ensemble of different conformations (e.g., inactive and active-like). In order to predict reliable inactive state model, should an antagonist or inverse agonist be added to the receptor for simulations, similar for the other three modeled class A GPCRs?

2. In the presented free energy profiles of GPR88, b2AR, M2R and uOR, there is consistently an energy minimum at ~ 4 angstrom TM3-TM6 distance, which is even shorter than the distance of the inactive state models. What is this conformation? Does it have any biological meaning or perhaps result from simulation artificial effects?

3. It has been shown that error analysis could be carried out for metadynamics simulations. Could error bars be added to the free energy profiles? This will be especially useful to evaluate significance of the shallow low energy wells in the free energy profile of GPR88 (Fig. 5A).

4. Why there is no clear energy minimum for the activate state in free energy profiles of the M2R and uOR? They seem different from findings of the other two receptors.

5. Previous studies showed that microsecond-timescale direct MD simulation was able to model deactivation of GPCRs, notably b2AR. Could direct MD simulations or enhanced sampling simulations without using collective variables be performed to predict probably more accurate inactive state models of the GPCRs, rather than using metadynamics that requires biasing along selected collective variables (only TM3-TM6 distance in case of the current study)? Note that there could be large/slow conformational changes in other regions of GPCRs, such as TM7, TM5 and water-mediated hydrogen-bonding network as shown by the authors.

Reviewer #3 (Remarks to the Author):

This manuscript by Chen et al reports the structural characterization of a signaling complex of GPR88 in complex with a synthetic agonist, 2-PCCA. GPR88 is an understudied orphan GPCR in the brain with potential functions in regulating reward and cognition. The endogenous ligand of GPR88 has not been found yet. The authors first demonstrated that GPR88 mainly signals through Gi other than Gs, Gq, and G13, and then obtained a high-resolution cryo-EM structure of GPR88 with Gi and 2-PCCA. A novel finding from this study is the identification of two binding sites for 2-PCCA and the allosteric cooperativity of ligand binding at these two sites. To further investigate receptor activation, the authors performed metadynamics simulations to model the inactive conformation of GPR88. In addition, the structural analysis revealed a potentially distinct water-mediated polar network in GPR88 involving a new conformation of the conserved NPxxY motif and the non-conserved residue H131. The unique structural properties of GPR88 and the non-conserved activation mechanism are quite interesting. The structural findings are supported by extensive mutagenesis data. Overall, this paper represents high-quality research in GPCR structural biology and pharmacology.

Some minor comments:

1. It is not clear whether the novel features observed in the 2-PCCA-GPR88-Gi structure including the two-site ligand binding, the disordered extracellular loops and the short TM6 of GPR88, and the unique intracellular water-mediated polar network are specifically associated with the synthetic agonist. This may comprise the impact of such an interesting structure. For example, 2-PCCA binds to the allosteric site located between the cytoplasmic ends of TM5 and TM6. Is this site specific to 2-PCCA or it can also accommodate other agonists including potential endogenous ligands? In particular, 2-PCCA seems to directly interact with Gi, which has never been observed for other GPCRs. Is it just a coincidence or it has certain physiological significance? Also, is the unique active conformation of TM6 (Fig. S4A) caused by the binding of 2-PCCA, or is it an inherent feature of GPR88? It is difficult to address those questions since the endogenous ligands of GPR88 are not clear and there are not many synthetic ligands developed for GPR88. However, it will be helpful to acknowledge such potential limitation of the study.

2. For the disordered extracellular region including N-terminus, ECL1, and ECL2, could it be caused by the N-terminal BRIL insertion? Since the conserved CysECL2-CysTM3 disulfide bond does not exist in GPR88, it is possible that ECL1, ECL2 and the N-terminal region pack against each other to form an ordered structure (similar to rhodopsin), and the extra N-terminal BRIL may break it. The authors may compare the basal activity of wtGPR88 and N-BRIL-GPR88 to see if

the N-terminal BRIL insertion affects GPR88 signaling, or at least acknowledge this possibility.

3. The authors discussed several non-conserved structural motifs in GPR88 and further claimed that "GPR88 may signal through a distinctive molecular mechanism, consistent with the different micro-switches observed in GPR88." (line 332) and that "These divergent features in the G protein-coupling interface, together with a set of non-conserved micro-switches in the transmembrane core, indicate

that GPR88 may belong to a unique subfamily of class A GPCRs."(line 421). This may be overreaching. For example, W6.48 is only conserved in ~70% of Class A GPCRs and P5.50 is not conserved in many lipid GPCRs including S1p receptors, LPA receptors, and cannabinoid receptors. Also, for the muscarinic receptor M2R, even though the PIF motif is conserved, it didn't play an important role in the receptor activation (doi:10.1038/nature12735). In fact, there are more than 40 GPCRs (if remember correctly) with solved active structures and many of them don't have conserved micro-switches. Every Class A GPCR has its specific features in G protein coupling and receptor activation more or less.

4. On a related topic, it will be helpful to exam if Q5.50 and H3.43 also exist in other GPCRs. Maybe GPR88 is structurally related to certain Class A GPCRs even though they don't share high sequence similarities.

5. The authors need to indicate the PDB IDs of structures used in their structural comparison analysis in all figure legends.

6. Figure 4 is all about the validation of their simulations protocol and has nothing to do with GPR88. It should be moved to supplementary information or combined with Figure 5.

7. In the abstract, the authors claimed that their study will facilitate the de-orphanization of GPR88. However, there is very little discussion of potential ligands of GPR88 other than a simple comparison with an S1pR. It will be interesting to provide more discussion. For example, if the endogenous ligand is a lipid, based on the space and the shape of the orthosteric pocket, is it a long-chain or a short-chain lipid? Is it linear, branched, or with some cyclic structure? What is the electrostatic charge potential of the orthosteric pocket? Can it accommodate any residues from ECL2?

-CZ

We thank all reviewers for their constructive and helpful comments. Please see our detailed responses to the comments below. The reviewers' comments are in black font and our responses are in blue font.

REVIEWER COMMENTS

Reviewer #1 (Remarks to the Author):

The manuscript entitled "Structure of the orphan GPR88-Gi complex reveals unique ligand binding and signaling mechanism" by Chen and colleagues is well written and clear. The authors describe the cryoEM structure of a complex including the orphan G protein coupled receptor (GPCR) GPR88, the Gi heterotrimer, and the recently discovered synthetic ligand (1R, 2R)-2-PCCA. Important features described in the present paper include the identification of 2 ligand binding pockets that intriguingly bind 2 molecules of 2-PCCA. Moreover, for the first time it is shown how the $\alpha 5$ helix in the Gai1 C-terminus contributes to one of the pockets highlighting a novel mechanism of G protein activation. The paper is of high importance as orphan receptors are untapped pharmacological targets with relevant physiological roles and this is especially true for GPR88. At the same time, the only orphan GPCR structure currently available is that of GPR52, thus new information related to other orphan GPCRs may guide deorphanization efforts leading to new insights into orphan receptor biology.

Thanks for the reviewer's positive comments on this study.

Comments:

The orthosteric binding site is defined as the site of interaction with endogenous ligands, while molecules that bind to the allosteric site normally only modulate signals initiated by agonists binding to the orthosteric site, without having any intrinsic signaling property. In the manuscript, this terminology is used as a proxy for extracellular binding pocket (orthosteric) or intramembrane binding pocket (allosteric). Considering that we do not know where the endogenous ligand binds in GPR88 (it could be one site, the other, or both), this terminology seems to be inappropriate especially in a case like the one shown here, where the same ligand binds to both sites. Experiments where point mutations in the proposed allosteric pocket (L2095.55, V2165.62, V2195.65 and L2876.38 to alanine) reduced the EC₅₀ without affecting GPR88 membrane targeting and E_{max} (Fig. 2I-K, and S3A) suggest it could be an allosteric site. However, mutations V216F and G283V (that do not affect GPR88 membrane localization) completely eliminate Gi signaling supporting an active role of this "allosteric" site in the generation of GPR88 signal. The high constitutive activity shown for G283V does not represent an appropriate control as signal can be activated by intracellular locations.

Moreover, loss of Gi activation by 2-PCCA due to mutations in the “orthosteric” site (for example: W3227.39 and W842.56 to smaller alkyl residues) are inconclusive, as a >90% reduction of GPR88 is observed at the plasma membrane (it seems like there is a strong correlation between GPR88 membrane targeting and Emax in Fig. 2I-J). Competition assays using a different agonist that do not bind to both pockets would help to establish which one is the actual orthosteric site. If sites are named because of structural homology with other GPCRs, this should be made clear in the manuscript.

We completely agree with the reviewer’s comment on the terminology of orthosteric and allosteric binding site. Based on the available functional data and lack of enough ligands to be tested on GPR88, we cannot exactly specify these two binding sites orthosterically or allosterically for sure. We named the two sites by comparing with other class A GPCR structures, an orthosteric site exposed to extracellular surface, and allosteric site(s) within intramembrane regions observed in other structures containing allosteric ligands (as indicated in ref 18-23 in the maintext). Therefore, we have deleted the usage of “allosteric” in the abstract and added some statement for clarification in page 5 (line 135-137).

As for the G283V mutation, we agree with the reviewer that it supports the activity of the “allosteric site”, but more specifically, it suggest the functional role of the “allosteric site” for binding of 2-PCCA. Therefore, we believe the data that showing the unaffected receptor expression level and basal activity is needed to support that the loss of function for 2-PCCA in the G283V mutant is not due to the loss of function of the receptor itself, instead, it’s because of the impaired binding for 2-PCCA.

For the loss of function of W3227.39 and W842.56, we actually compared the Gi activity to WT receptor that are normalized to the same expression level (through plasmid titration) as these mutants, as shown in Fig. 2 i-k, for example, the WT(1:32) titration show similar expression level as these mutatants.

Statistical analysis applied to the data shown in Fig. 1C, Fig 2I-K, Fig S1B, and Fig. S5C-E is not reported anywhere in the manuscript.

We added statistical values in the revised Fig. 1c, Fig. 2i-k and Supplementary Fig. 5c-e. As stated in the legend, we performed the statistical analysis using the ordinary one-way ANOVA followed by the Dunnett’s post hoc test with the expression-matched (colored) WT response. Supplementary Fig. 1b was not intended to demonstrate statistical significance between the GPCR constructs and thus we do not add statistical values in the graph panel.

In the discussion it is speculated that a lipid may be the endogenous ligand of GPR88. 2-PCCA interacts with GPR88 in both the identified pockets. Based on

these information, could it be speculated that the 2 binding sites bind an endogenous ligand that interacts with both orthosteric and allosteric pockets simultaneously? Possibly a dimer?

This is an interesting question. Based on our experimental results, we did expect that the endogenous ligand of GPR88 may also bind to both the orthosteric and allosteric pockets (2 ligand, 1 receptor mode). However, since the orthosteric pocket is at the extracellular side and the allosteric pocket is at the intracellular side with rotated position, it is hard to speculate how the endogenous ligand interacts with the two sites simultaneously and functions as a dimer.

The observation that the allosteric pocket is formed by GPR88 and Gi1 raises a question about the possible contribution of other G alpha proteins (Gi2, Gi3, Go, and Gz) in the formation of this binding pocket. Are the amino acids involved in the interaction conserved among other G proteins? Or this configuration is possible only with Gi1 creating an intrinsic bias in GPR88 activation towards this G protein?

We appreciate the reviewer's suggestion. We performed sequence alignment at both the N-term and C-term interaction sites between GPR88 and G proteins shown as below. The amino acids involved in the interaction are quite conserved in different G proteins (indicated by the black arrowhead), especially at the C-term helix that forms the allosteric pocket with GPR88. So, it is possible that this allosteric binding pocket still exists when GPR88 binds to other G protein alpha subunits.

		N-term interaction sites			C-term interaction sites		
		▼	▼		▼	▼	▼
Gi1	24	EDGEKAAAREVKLL	VFDAVTDV	IIKNNLKDCGL	F	354
Gi2	24	EDGEKAAAREVKLL	VFDAVTDV	IIKNNLKDCGL	F	355
Gi3	24	EDGEKAAKEVKLL	VFDAVTDV	IIKNNLKECGL	Y	354
Go	24	EDGISAAKDVKLL	VFDAVTD	IIIANNLKRGCG	LY	354
Gz	24	SESQRQRREIKLL	VFDAVTDV	IIQNNLKYIGL	C	355

Minor comments:

Line 91-93: Please consider including references to the studies demonstrating Gi/o coupling of GPR88. -Added

Line 103: Reference to Fig. 1A is missing. -Added

Lines 121-126: It is not clear why GPR88 and GPR52 should have similarities in the ECL2 structure and function, besides the fact that they are both class A GPCRs and orphans. Is there a high homology of amino acid sequence in this

region between GPR88 and GPR52? A clarification would be helpful.

Thank the reviewer for raising this point. GPR52 is the first reported self-activated class A GPCR and its basal activity is quite high. Considering high overall structural similarity among class A GPCRs and the similar high basal activity for these two GPCRs, we speculate that GPR88 might bear the same mechanism of activation relative to GPR52. However, the sequence alignment of ECL2 of GPR52 and GPR88 shows low homology as below (the key residues for GPR52 activation indicated by black arrowhead), so it is possible that GPR88 utilizes a distinct self-activating mechanism. We added this clarification in page5 (ling 125-127) of the revised manuscript and added the sequence alignment results in the revised Supplementary Fig. 2b.

```

          ▼      ▼      ▼      ▼
GPR52 ECL2  - - GWG - KPGYHGD I FEWCAT SWL T S  179-200
GPR88 ECL2  LPPWAPRPG - - - - - - - - AAPP RVI - 176-190

```

Line 213: mutant P177G is reported as P178G in Figure 2I and Supplementary Figure S3A.

It is P178G, we've corrected it in Fig. 2c-d and in line 216.

Line 327: Please consider including references supporting this statement. Added in the revised manuscript line 332.

Line 376-378: This is not completely true, as mutation H131A shows a drastic reduction in membrane targeting (>60%), but similar, if not better, E_{max} compared to WT (Fig. S5B-E).

Thank the reviewer for raising this point. We have deleted the sentence in the text to avoid overstatement.

Methods: The origin of the DNA constructs used in the study (GPR88, Gi1, scFv16, etc.) should be reported.

We have added these information in the methods part.

Reviewer #2 (Remarks to the Author):

In this manuscript, cryo-EM structure was determined for orphan GPR88 in complex with a synthetic agonist 2-PCCR and the Gi1 protein. Two ligand molecules bound simultaneously to the orthosteric site and a newly identified allosteric binding site formed by the TM5 and TM6 intracellular domains and the C-terminus of the Gi1 $\alpha 5$ helix. Notably, metadynamics simulations were performed to predict an inactive state model of GPR88.

Overall, the structure is very interesting and the manuscript is well written. However, a number of major suggestions that may help improving the

manuscript include:

1. The authors mentioned that GPR88 exhibits high basal activity, suggesting that the apo receptor likely samples an ensemble of different conformations (e.g., inactive and active-like). In order to predict reliable inactive state model, should an antagonist or inverse agonist be added to the receptor for simulations, similar for the other three modeled class A GPCRs?

To the best of our knowledge, antagonists for GPR88 have not yet been described, as shown by our literature studies and searches in databases such as IUPHAR/BPS Guide to Pharmacology, UniProt or ChEMBL. Our inactive-state modeling for GPR88 and the other three reference GPCRs has been done in the absence of an antagonist. All predicted inactive models show an inward shifted TM6 and outward shifted TM7, closing the G protein binding site and thus indicate inactive properties. These structures are clearly inactive and have been obtained without having to consider the uncertainties in ligand classification, especially for neutral antagonists / inverse agonists.

2. In the presented free energy profiles of GPR88, b2AR, M2R and uOR, there is consistently an energy minimum at ~4 angstrom TM3-TM6 distance, which is even shorter than the distance of the inactive state models. What is this conformation? Does it have any biological meaning or perhaps result from simulation artificial effects?

We thank the reviewer for this thoughtful comment. We have added the following comment to the maintext on pages 12-13 (line 361-371) : “Notably, a local minimum is observable around 4 Å in all simulation systems. The receptor conformation in this minimum resembles the global minimum model except that the TM3-TM6 distance is shorter (caused by a slight inward shift of TM3 towards TM6). The meaning of this receptor conformation is not clear, but it might represent an alternative inactive conformation. The existence of alternative inactive conformations has also been suggested in previous studies that applied MD simulations but also by NMR and DEER spectroscopy experiments”.

3. It has been shown that error analysis could be carried out for metadynamics simulations. Could error bars be added to the free energy profiles? This will be especially useful to evaluate significance of the shallow low energy wells in the free energy profile of GPR88 (Fig. 5A).

Great suggestion. Error analyses could be performed for all receptors according to the PLUMED Master ISDD tutorial 2020. We have added error bars to all free-energy profiles of the revised Supplementary Fig. 6a-d. Interestingly, the error bars show that the global minima are favored compared to other local

minima. Supplementary Fig. 6a-d and its figure legend have been modified accordingly: "Error bars are indicated in gray and show that the global minima are more favorable compared to other local minima."

4. Why there is no clear energy minimum for the activate state in free energy profiles of the M2R and uOR? They seem different from findings of the other two receptors.

Indeed, for M2R and mOR, we did not observe a significant local minimum that corresponds to an active state without the G protein. This is likely a real difference in the conformational behavior between the receptors.

5. Previous studies showed that microsecond-timescale direct MD simulation was able to model deactivation of GPCRs, notably b2AR. Could direct MD simulations or enhanced sampling simulations without using collective variables be performed to predict probably more accurate inactive state models of the GPCRs, rather than using metadynamics that requires biasing along selected collective variables (only TM3-TM6 distance in case of the current study)? Note that there could be large/slow conformational changes in other regions of GPCRs, such as TM7, TM5 and water-mediated hydrogen-bonding network as shown by the authors.

Thanks for the reviewer's comments. Very long MD simulations can also give inactive conformations. Because of the very slow relaxation mechanisms mentioned by the referee, they may move, in some cases, to long-lived secondary minima, rather than the globally most stable inactive state. One major advantage of the enhanced sampling simulations is that we can apply error analyses and convergence tests that are not available for unconstrained simulations. The protocol described provides a practical, accessible and above all receptor-independent access to conformations of different activity. Furthermore, it is well suited to massively parallel hardware. We regard metadynamics as a very promising technique to simulate conformational changes of receptors. MD simulations as an alternative approach have been mentioned now on page 12 (line 342-344) of the manuscript: "This method is an attractive alternative to long-term unbiased MD simulations to investigate on conformational changes of GPCRs".

Reviewer #3 (Remarks to the Author):

This manuscript by Chen et al reports the structural characterization of a signaling complex of GPR88 in complex with a synthetic agonist, 2-PCCA. GPR88 is an understudied orphan GPCR in the brain with potential functions in regulating reward and cognition. The endogenous ligand of GPR88 has not

been found yet. The authors first demonstrated that GPR88 mainly signals through Gi other than Gs, Gq, and G13, and then obtained a high-resolution cryo-EM structure of GPR88 with Gi and 2-PCCA. A novel finding from this study is the identification of two binding sites for 2-PCCA and the allosteric cooperativity of ligand binding at these two sites. To further investigate receptor activation, the authors performed metadynamics simulations to model the inactive conformation of GPR88. In addition, the structural analysis revealed a potentially distinct water-mediated polar network in GPR88 involving a new conformation of the conserved NPxxY motif and the non-conserved residue H131. The unique structural properties of GPR88 and the non-conserved activation mechanism are quite interesting. The structural findings are supported by extensive mutagenesis data. Overall, this paper represents high-quality research in GPCR structural biology and pharmacology.

Thanks for the reviewer's positive comments on this study.

Some minor comments:

1. It is not clear whether the novel features observed in the 2-PCCA-GPR88-Gi structure including the two-site ligand binding, the disordered extracellular loops and the short TM6 of GPR88, and the unique intracellular water-mediated polar network are specifically associated with the synthetic agonist. This may comprise the impact of such an interesting structure. For example, 2-PCCA binds to the allosteric site located between the cytoplasmic ends of TM5 and TM6. Is this site specific to 2-PCCA or it can also accommodate other agonists including potential endogenous ligands? In particular, 2-PCCA seems to directly interact with Gi, which has never been observed for other GPCRs. Is it just a coincidence or it has certain physiological significance? Also, is the unique active conformation of TM6 (Fig. S4A) caused by the binding of 2-PCCA, or is it an inherent feature of GPR88? It is difficult to address those questions since the endogenous ligands of GPR88 are not clear and there are not many synthetic ligands developed for GPR88. However, it will be helpful to acknowledge such potential limitation of the study.

Thank the reviewer for sharing the insightful thoughts. We are not able to conclude the novel structural features observed are specifically associated with the 2-PCCA, considering the unknown endogenous ligand and lack of enough ligands on GPR88. Therefore, we cannot exclude the several possibilities that the reviewer mentioned for sure. For example, we admit that it could be a coincidence that 2-PCCA directly interacts with Gi. We added some discussion to acknowledge such limitations in the DISCUSSION section (page 16, line 466-469).

2. For the disordered extracellular region including N-terminus, ECL1, and

ECL2, could it be caused by the N-terminal BRIL insertion? Since the conserved CysECL2-CysTM3 disulfide bond does not exist in GPR88, it is possible that ECL1, ECL2 and the N-terminal region pack against each other to form an ordered structure (similar to rhodopsin), and the extra N-terminal BRIL may break it. The authors may compare the basal activity of wtGPR88 and N-BRIL-GPR88 to see if the N-terminal BRIL insertion affects GPR88 signaling, or at least acknowledge this possibility.

Thank the reviewer for raising this point. This N-terminal fusion strategy is widely used in enhancing GPCR recombinant expression and facilitating crystal packing. Previous reported crystal structures of N-terminal GPCR fusion does not show noticeable difference relative to the native receptor structure and activity, while we indeed acknowledge the possibility that the N-BRIL fusion might somehow affect GPR88 signaling. We have added some statement on page 27.

3. The authors discussed several non-conserved structural motifs in GPR88 and further claimed that "GPR88 may signal through a distinctive molecular mechanism, consistent with the different micro-switches observed in GPR88." (line 332) and that "These divergent features in the G protein-coupling interface, together with a set of non-conserved micro-switches in the transmembrane core, indicate that GPR88 may belong to a unique subfamily of class A GPCRs."(line 421). This may be overreaching. For example, W6.48 is only conserved in ~70% of Class A GPCRs and P5.50 is not conserved in many lipid GPCRs including S1p receptors, LPA receptors, and cannabinoid receptors. Also, for the muscarinic receptor M2R, even though the PIF motif is conserved, it didn't play an important role in the receptor activation (doi:10.1038/nature12735). In fact, there are more than 40 GPCRs (if remember correctly) with solved active structures and many of them don't have conserved micro-switches. Every Class A GPCR has its specific features in G protein coupling and receptor activation more or less.

We totally agree with the reviewer's comment in this regard. Considering the multiple points mentioned, we may overstate that GPR88 belongs to a unique subfamily of class A GPCRs. We have rephrased it to "These divergent features in the G-protein-coupling interface, together with a set of non-conserved micro-switches in the transmembrane core, further suggest that GPR88 may utilize a different mechanism for signaling transduction." in page 15 (line 434-437).

4. On a related topic, it will be helpful to exam if Q5.50 and H3.43 also exist in other GPCRs. Maybe GPRR88 is structurally related to certain Class A GPCRs even though they don't share high sequence similarities.

We appreciate this useful suggestion. Base on the generic number tables of

class A GPCRs on the GPCR database website (<https://gpcrdb.org/residue/residuetable>), the conserved P5.50 has a few variants like L5.50 (CB1 and CB2 receptor), A5.50(GPBA receptor), T5.50 (LPA1 receptor), M5.50 (MC4 receptor), I5.50(S1PR3), S5.50(TP receptor), V5.50(PAF receptor) and N5.50(P2YR12). But we didn't find the exact Q5.50 in this position. In addition, H3.43 is in the same situation. This position, which is normally a leucine, could be a methionine (PAR1 and PAR2) or small amino acids with hydrophobic side chains such as valine (H1 receptor), alanine (EP3 receptor) and isoleucine (ghrelin receptor). However, in human class A GPCRs, there is no H3.43 observed before. Thus, we may speculate that the interaction of Q5.50 and H3.43 is a unique feature for this orphan receptor GPR88.

5. The authors need to indicate the PDB IDs of structures used in their structural comparison analysis in all figure legends.

We have added PDB IDs in all figures.

6. Figure 4 is all about the validation of their simulations protocol and has nothing to do with GPR88. It should be moved to supplementary information or combined with Figure 5.

We agree this point and have followed this suggestion. The validation of simulation protocols and comparison of active and calculated inactive structures for β 2AR, M2R and μ OR was moved to the new Supplementary Fig. 6. The captions, figure numbering and the main text have been modified accordingly.

7. In the abstract, the authors claimed that their study will facilitate the de-orphanization of GPR88. However, there is very little discussion of potential ligands of GPR88 other than a simple comparison with an S1pR. It will be interesting to provide more discussion. For example, if the endogenous ligand is a lipid, based on the space and the shape of the orthosteric pocket, is it a long-chain or a short-chain lipid? Is it linear, branched, or with some cyclic structure? What is the electrostatic charge potential of the orthosteric pocket? Can it accommodate any residues from ECL2?

Thank the reviewer for this comment. While we are not able to de-orphanize GPR88 simply based on this agonist-bound structure, there are some estimations in terms of binding modes for endogenous lipid binding by comparing with that of 2-PCCA. We agree with the reviewer that it is of interest to think more about the potential shape of the endogeneous ligand. According to the current data, we suspect that the endogenous ligand might be a long-chain (~14-16C) lipid with branched structure. However, we think the ligand diversity of GPR88 is not well investigated yet, and the current structure only

provides limited information. Structural studies of GPCRs have shown that one GPCR can bind to a diversity of ligand with different chemotypes (e.g. the beta2AR, whose endogeneous ligand is very small norepinephrine, while some of the ligands are very long and can extended to other sites). Therefore, we think it might be overreached to say more details about the shape of the endogenous ligand in the current study just based on the 2-PCCA binding pocket.

We generate the electrostatic change potential map of the orthosteric pocket below, which is quite neutral. Actually most of the residues in the missing ECL2 are small hydrophobic residues (Supplementary Figure 1), it's possible that the ECL2 may insert into the pocket for receptor self-activating. We discussed this possibility in page 5 (line 122-124).

REVIEWERS' COMMENTS

Reviewer #1 (Remarks to the Author):

The authors answered most of my concerns even though I believe they should still address the few minor points raised below.

The way statistical analysis was performed is now reported in the figure legends, however, the results of this analysis are not indicated on the related figures.

"We performed sequence alignment at both the N-term and C-term interaction sites between GPR88 and G proteins shown as below. The amino acids involved in the interaction are quite conserved in different G proteins (indicated by the black arrowhead), especially at the C-term helix that forms the allosteric pocket with GPR88. So, it is possible that this allosteric binding pocket still exists when GPR88 binds to other G protein alpha subunits."

This analysis and the related discussion have not been inserted in the manuscript.

Methods: The origin of the DNA constructs used in the study (GPR88, Gi1,scFv16, etc.) should still be reported. Where did the authors obtained the original plasmids containing the coding sequences used to subclone the constructs used in the present study?

"Line 327: Please consider including references supporting this statement. Added in the revised manuscript line 332." High constitutive activity of GPR88 was also previously reported and should be referenced.

Supplementary Figure 2 reports the density maps of N-terminal and C-terminal α helices of Gai1 (α N and α 5) and not Gas.

Reviewer #2 (Remarks to the Author):

The authors have properly addressed my comments in the revised manuscript.

Reviewer #3 (Remarks to the Author):

The authors have addressed all my concerns. The revised manuscript is well written. This new GPCR structure will be of great interest to the fields of GPCR structural biology and pharmacology.

We thank all the reviewers for their positive comments. In the last round of revision, there were few points raised by the reviewer #1, and we fully addressed and provided our point-by-point responses (see next page below). After that, our manuscript has been **accepted in principle** ready for galley proofing. However, during that period of time, we solved a new CryoEM structure of the GPR88-Gi complex in apo state at 3.0 Å. Unexpectedly, we also found a similar electron density within the orthosteric binding pocket in the apo GPR88 structure, compared to the synthetic agonist 2-PCCA-bound structure. That is, the electron density suggested an endogenous ligand is likely bound during the receptor purification, instead of the orthosteric 2-PCCA we were claiming. Other than this point, the main conclusions in the current manuscript, especially regarding newly discovered allosteric binding site, remain the same. We believe even the result of dual binding sites of 2-PCCA is changed, the two CryoEM structures (apo and 2-PCCA-bound) of the orphan GPR88-Gi complex represent important and novel insights into GPCR field. After seeking the editor's advice, we have incorporated the apo GPR88-Gi structure and revised the manuscript for resubmission.

The revised points are:

- (1) We labeled all the new edits in the updated manuscript in **blue** font and prepared a merged pdf of "**Revised manuscript**" including new main figures for additional review purpose;
- (2) We modified main figures Fig. 1, Fig. 2 and Fig.5 by incorporating the new apo GPR88-Gi structure;
- (3) We provided "**Revised supplementary figures**" (order changed, new figures added and reorganized) with updated Table S1, and the PDB validation reports for the apo and 2-PCCA-bound GPR88-Gi structure.

We really apologize for any inconvenience and delay that may cause. We are much grateful for the reviewers' precious time to review this updated manuscript.

The point-by-point responses to the reviewer's comments in last round of revision is as below. The reviewer's comments are in black font and our responses are in blue font.

REVIEWER COMMENTS

Reviewer #1 (Remarks to the Author):

The authors answered most of my concerns even though I believe they should still address the few minor points raised below.

Thank the reviewer for the positive comments.

The way statistical analysis was performed is now reported in the figure legends, however, the results of this analysis are not indicated on the related figures.

As requested by the reviewer, the results of the statistic analysis are now incorporated in Fig. 1c, Fig.2-i-k and supplementary Fig. 5c-e.

"We performed sequence alignment at both the N-term and C-term interaction sites between GPR88 and G proteins shown as below. The amino acids involved in the interaction are quite conserved in different G proteins (indicated by the black arrowhead), especially at the C-term helix that forms the allosteric pocket with GPR88. So, it is possible that this allosteric binding pocket still exists when GPR88 binds to other G protein alpha subunits."

This analysis and the related discussion have not been inserted in the manuscript.

We have inserted the discussion in line 414-419. The sequence alignment are inserted into Supplementary Fig. 8a. The figure legends and the main text have been modified accordingly.

Methods: The origin of the DNA constructs used in the study (GPR88, Gi1,scFv16, etc.) should still be reported. Where did the authors obtained the original plasmids containing the coding sequences used to subclone the constructs used in the present study?

We have added the origin of the DNA coding sequences in the "Method" section- Expression and Purification of GPR88.

"Line 327: Please consider including references supporting this statement. Added in the revised manuscript line 332." High constitutive activity of GPR88 was also previously reported and should be referenced.

We have added the references in line 332. We also refer to the data in the supplementary Fig. 1b.

Supplementary Figure 2 reports the density maps of N-terminal and C-terminal α helices of Gai1 (α N and α 5) and not Gas.

We have corrected the mistake.

REVIEWERS' COMMENTS

Reviewer #1 (Remarks to the Author):

My concerns have been addressed and I believe the manuscript has been greatly improved.

Reviewer #2 (Remarks to the Author):

Since all ligands were removed in the simulations, changes in the newly presented structures do not affect the simulations.

However, it's unclear why coordinates could be assigned for 2-PCCA on the protein surface, but not for the "putative endogenous agonist" that binds in the orthosteric pocket within the receptor transmembrane domain. Should the latter be less flexible and thus easier for mapping the coordinates?

Reviewer #3 (Remarks to the Author):

The authors included a new apo structure in the revised manuscript. Clearly, the new structural analysis led the authors to conclude that 2-PCCA only occupies the allosteric site but not the orthosteric site, in contrast to their previous claim of two-site binding of 2-PCCA. This may serve as a very good example of how modeling GPCR ligands to cryo-EM maps can be ambiguous and not reliable, even though the overall resolution of the 2-PCCA-GPR88-Gi is high enough ($\sim 2.4\text{\AA}$) to show densities of water molecules. I highly suggest that the authors include such experience that how they came up with a two-site binding hypothesis based on one high-resolution structure but later overthrew it with more experimental evidence in their Discussion. This will provide highly valuable information in the GPCR structural biology field and warn other researchers about the overinterpretation of their cryo-EM maps.

I don't have major problems with the new structure and the revised discussion of their results. I believe this is a very interesting piece of work in GPCR structural biology with a high impact. Several minor comments:

1. The GTPase Glo assay is not a reliable assay for measuring GPCR basal activities. Many GPCR exhibit much higher basal activities in detergent buffer compared to them in lipid environments. This is not a big problem though since the authors also did TGF α shedding assays to prove the high basal activity.

2. The orthosteric pocket is highly open to the extracellular milieu, which is not common in lipid GPCRs. It may be a peptide or even a detergent molecule. It will be helpful to show the detailed environment of the orthosteric pocket, e.g. is it highly hydrophobic or hydrophilic?

We thank all the reviewers for their positive comments. Please see our detailed responses to the comments below. The reviewers' comments are in black font and our responses are in blue font.

Reviewer #1 (Remarks to the Author):

My concerns have been addressed and I believe the manuscript has been greatly improved.

We thank the reviewer for the positive comments.

Reviewer #2 (Remarks to the Author):

Since all ligands were removed in the simulations, changes in the newly presented structures do not affect the simulations.

However, it's unclear why coordinates could be assigned for 2-PCCA on the protein surface, but not for the "putative endogenous agonist" that binds in the orthosteric pocket within the receptor transmembrane domain. Should the latter be less flexible and thus easier for mapping the coordinates?

As shown in the Fig. 1, we now have another structure ("apo-state") where the 2-PCCA was NOT added during sample preparation, this structure therefore could serve as a control for us to assign the ligand density. As discussed in the paper, we observed a similar density in the orthosteric pocket but no density at the protein surface (or the allosteric site) in 'apo' structure. In other words, the density at the allosteric site was only found in the sample with 2-PCCA. In addition, the density in the allosteric site fit very-well with the synthetic ligand structure and actually form extensive interactions with the receptor. Therefore, we are quite confident that the density at the allosteric site should correspond to 2-PCCA. However, we are not sure whether the density located in the orthosteric pocket is the "putative endogenous agonist" or 2-PCCA because we found this density in both maps/samples.

Reviewer #3 (Remarks to the Author):

The authors included a new apo structure in the revised manuscript. Clearly, the new structural analysis led the authors to conclude that 2-PCCA only occupies the allosteric site but not the orthosteric site, in contrast to their previous claim of two-site binding of 2-PCCA. This may serve as a very good example of how modeling GPCR ligands to cryo-EM maps can be ambiguous and not reliable, even though the overall resolution of the 2-PCCA-GPR88-Gi is high enough (~2.4Å) to show densities of water molecules. I highly suggested that the authors include such experience that how they came up with a two-site binding hypothesis based on one high-resolution structure but later overthrew it with more experimental evidence in their Discussion.

This will provide highly valuable information in the GPCR structural biology field and warn other researchers about the overinterpretation of their cryo-EM maps.

We thank the reviewer's suggestion and have added the discussion starting at line 457 as following: "The high-resolution map of 2-PCCA-bound GPR88-Gi1 complex led us to the initial assumption that this synthetic agonist unambiguously occupies in both orthosteric and allosteric pockets. However, it was greatly challenged since a clear ligand density still exists in the orthosteric pocket of the apo-GPR88-Gi cryo-EM map as a control. The comparison of two maps suggested that a putative endogenous ligand in the orthosteric pocket may be co-purified with GPR88. This could serve as a caveat on modeling GPCR ligands to cryo-EM maps and interpreting suspicious densities in the cryo-EM maps."

I don't have major problems with the new structure and the revised discussion of their results. I believe this is a very interesting piece of work in GPCR structural biology with a high impact. Several minor comments:

1. The GTPase Glo assay is not a reliable assay for measuring GPCR basal activities. Many GPCR exhibit much higher basal activities in detergent buffer compared to them in lipid environments. This is not a big problem though since the authors also did TGFa shedding assays to prove the high basal activity.

Thank the reviewer for pointing this out. We agree with the reviewer that the Glo-assay in detergent might be not that accurate, so we performed TGFa shedding assays for cross-validation. Both assays suggested the high-basal activity of GPR88.

2. The orthosteric pocket is highly open to the extracellular milieu, which is not common in lipid GPCRs. It may be a peptide or even a detergent molecule. It will be helpful to show the detailed environment of the orthosteric pocket, e.g. is it highly hydrophobic or hydrophilic?

We showed the environment of the orthosteric pocket in Fig. 2c and Supplementary Fig.5d. It shows that the pocket is highly hydrophobic in the TM tunnel while charged in the extracellular surface. Based on the shape of the density and such property of the pocket, we believe the density could represent a lipid-like molecule with its hydrophilic head lies in the extracellular surface while the hydrophobic chain inserts into the tunnel, similar to the binding pose of the modeled orthosteric 2-PCCA as shown in Supplementary Fig.5a-d.